# Deep Adaptive Semantic Logic (DASL): Compiling Declarative Knowledge into Deep Neural Networks

## Abstract

We introduce **Deep Adaptive Semantic Logic (DASL)**, a novel framework for automating the generation of deep neural networks that incorporates user-provided formal knowledge to improve learning from data. We provide formal semantics that demonstrate that our knowledge representation captures all of first order logic and that finite sampling from infinite domains converges to correct truth values. DASL's representation improves on prior neuro-symbolic work by avoiding vanishing gradients, allowing deeper logical structure, and enabling richer interactions between the knowledge and learning components. We illustrate DASL through a toy problem in which we add structure to an image classification task and demonstrate that knowledge of that structure reduces data requirements by a factor of 1000. We apply DASL on a visual relationship detection task and demonstrate that the addition of commonsense knowledge improves performance by 10.7% in conditions of data scarcity.

## 1 Introduction

Early work on Artificial Intelligence focused on Knowledge Representation and Reasoning (KRR) through the application of techniques from mathematical logic [Genesereth & Nilsson (1987)]. The compositionality of KRR techniques provides expressive power for capturing expert knowledge in the form of rules or assertions (*declarative knowledge*), but they are brittle and unable to generalize or scale. Recent work has focused on Deep Learning (DL), in which the parameters of complex functions are estimated from data [LeCun et al. (2015)]. DL techniques learn to recognize patterns not easily captured by rules and generalize well from data, but they often require large amounts of data for learning and in most cases do not reason at all [Yang et al. (2017); Garcez et al. (2012); Marcus (2018); Weiss et al. (2016)]. In this paper we present [**Deep Adaptive Semantic Logic (DASL)**], a framework that attempts to take advantage of the complementary strengths of KRR and DL by fitting a model simultaneously to data and declarative knowledge. DASL enables robust abstract reasoning and application of domain knowledge to reduce data requirements and control model generalization.

DASL represents declarative knowledge as assertions in first order logic. The relations and functions that make up the vocabulary of the domain are implemented by neural networks that can have arbitrary structure. The logical connectives in the assertions compose these networks into a single deep network that is trained to maximize their truth. Figure 1 provides an example network that implements a simple rule set through composition of network components performing image classification. Logical quantifiers "for all" and "there exists" generate subsamples of the data on which the network is trained. DASL treats labels like assertions about data, removing any distinction between knowledge and data. This provides a mechanism by which supervised, semi-supervised, unsupervised, and distantly supervised learning can take place simultaneously in a single network under a single training regime.

The field of neuro-symbolic computing [Garcez et al. (2019)] focuses on combining logical and neural network techniques in general, and the approach of [Serafini & Garcez (2016)] may be the closest of any prior work to DASL. To generate differentiable functions to support backpropagation, these approaches replace pure Boolean values of 0 and 1 for *True* and *False* with continuous values from $[0, 1]$ and select fuzzy logic operators for implementing the Boolean connectives. These operators generally employ maximum or minimum functions, removing all gradient information at the limits,

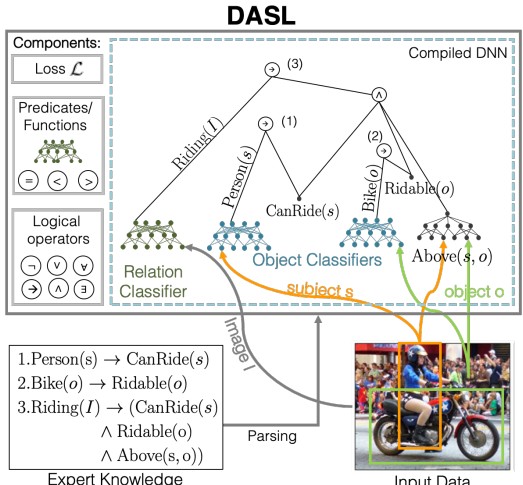

Figure 1: DASL integrates user-provided expert knowledge with training data to learn DNNs. It achieves this by compiling a DNN from knowledge, expressed in first order logic, and domain-specific neural components. This DNN is trained using backpropagation, fitting both the data and knowledge. Here DASL applies commonsense knowledge to the visual relationship detection task. $\wedge$ and $\rightarrow$ refer to 'and' and 'implies' connectives respectively.

or else they use a product, which drives derivatives toward $0$ so that there is very little gradient for learning (see subsection A.7). DASL circumvents these issues by using a logit representation of truth values, for which the range is all real numbers.

Approaches to knowledge representation, both in classical AI and in neuro-symbolic computing, often restrict the language to fragments of first order logic (FOL) in order to reduce computational complexity. We demonstrate that DASL captures full FOL with arbitrary nested quantifiers, function symbols, and equality by providing a single formal semantics that unifies DASL models with classical Tarski-style model theory [Chang & Keisler (1973)]. We show that DASL is sound and complete for full FOL. FOL requires infinite models in general, but we show that iterated finite sampling converges to correct truth values in the limit.

In this paper we show an application of DASL to learning from small amounts of data for two computer vision problems. The first problem is an illustrative toy problem based on the MNIST handwritten digit classification problem. The second is a well-known challenge problem of detecting visual relationships in images. In both cases, we demonstrate that the addition of declarative knowledge improves the performance of a vanilla DL model. This paper makes the following contributions:

1. The novel framework DASL, which compiles a network from declarative knowledge and bespoke domain-specific reusable component networks, enabling gradient-based learning of model components;

2. Grounding of the proposed framework in model theory, formally proving its soundness and completeness for full first order logic;

3. A logit representation of truth values that avoids vanishing gradients and allows deep logical structures for neural-symbolic systems;

4. Syntactic extensions that allow (i) restricted quantification over predicates and functions without violating first order logic constraints, and (ii) novel hybrid network architectures;

5. Evaluation on two computer vision problems with limited training data, demonstrating that knowledge reduces data requirements for learning deep models *e.g.* factor of 1000 for the MNIST toy problem and $10.7\%$ improvement in accuracy for visual relationship detection in conditions of data scarcity.

## 2 RELATED WORK

**Neuro-Symbolic Computing:** Early efforts to augment DNNs with logic focused on propositional logic, which supports only logical connectives between (atomic) propositions [Garcez et al. (2012; 2019)]. For example, KBANN [Towell & Shavlik (1994)] maps a set of propositions into a graph, constructs a neural network, and then trains it. DASL follows this basic idea but fully supports full first order logic (FOL) as well as arithmetic expressions.

Similar to several prior efforts [Hu et al. (2016); Rocktäschel et al. (2015); Li & Srikumar (2019)], DASL replaces Booleans with real-valued *pseudo-probabilities* to make the logical operations differentiable. This circumstance has motivated the invention of a collection of *ad hoc* aggregation operators for representing logical connectives [Detyniecki (2001)]. These include the *t-norm*, used by Logic Tensor Networks (LTNs) [Serafini & Garcez (2016)] and the above works. Instead, DASL uses a logit representation for truth values, which avoids vanishing gradients more comprehensively than the logarithmic representations of [Giannini et al. (2019); Rocktäschel et al. (2015)], enabling learning in deeper logical structures. DASL also differs in supporting multiple entity types, arithmetic, and non-traditional operations such as *softmax* that enable richer interaction between the NN and knowledge (Section 4).

Prior neuro-symbolic work has demonstrated the ability to represent predicate relations as neural networks, generally for formulas without any nested quantifiers. Full first-order logic allows arbitrary nesting of quantifiers and connectives, and allows function symbols and equality under "open world" semantics. DASL represents the first time that soundness and completeness have been established for a FOL system applied to neural networks, demonstrating that we can accommodate full first-order logic. There is also a body of work on tensor representations of logic over finite domains that is focused on efficient evaluation of formulas but is not concerned with learning [Sato (2017); Nguyen et al. (2018)].

**Compositional DL:** DASL is related to works that execute a task by composing trainable neural modules by parsing a query (in natural language) [Andreas et al. (2016); Yi et al. (2018a); Mao et al. (2019); Yi et al. (2018b)]. For example, [Yi et al. (2018b)] focuses on visual question answering and employs a differentiable tree-structured logic representation, similar to DASL. However, it only does so to learn to translate questions, whereas DASL learns the semantics of the application domain and can also integrate useful domain knowledge.

**Structured Learning:** Other work exploits underlying structure in the data or the label space to learn DNNs using techniques such conditional random fields, graph neural networks, attention models, etc., including [Belanger et al. (2017); Kim et al. (2017); Battaglia et al. (2018); Peng et al. (2018); Zheng et al. (2015)]. These methods impose structure by either adapting the DNN architecture Battaglia et al. (2018) or the loss function [Zheng et al. (2015)]. DASL instead imposes soft constraints by compiling DNNs based on rules that can be stated in a flexible manner using FOL.

**Semantic Reasoning:** By the *semantics* of a logical language we mean an interpretation of its symbols (which do not include logical connectives and quantifiers); a *model* in the sense of model theory [Weiss & D'Mello (1997)]. In common with several methods [Xie et al. (2019)], DASL grounds its entities in vector spaces (embeddings) and its predicates and functions in trainable modules. DASL builds on prior works on semantic representation techniques Pennington et al. (2014); Mikolov et al. (2013); Deerwester et al. (1990) by enabling logical statements to modify the entity embeddings so as to mirror semantic similarity in the application. Another research direction [Rocktäschel & Riedel (2017); Cohen et al. (2017); de Jong & Sha (2019)] is to structure DNNs and embedding spaces around proof trees to reduce the brittleness of theorem provers [Siekmann & Wrightson (1983)].

An alternative to representing predicates by networks is to restrict oneself to propositional knowledge [Xu et al. (2018)] which can be applied to the outputs of a neural network. Propositional logic cannot represent the rules in Figure 1, for example, but can be useful for placing restrictions on fixed outputs.

**Bayesian Belief Networks:** Substitution of pseudo-probabilities for Booleans fails to capture uncertainty the way fully Bayesian methods do [Jaynes (2003)]. Bayesian Belief networks [Pearl (2009)] accurately represent probabilities but lack expressivity and face computability challenges. Bayes nets are most comfortably confined to propositional logic. Efforts to extend them to first-order logic

include Markov Logic Networks [Richardson & Domingos (2006)], which use an undirected network to represent a distribution over a set of models, *i.e.*, *groundings* or *worlds* that can interpret a theory. The lifted inference approach [Kimmig et al. (2004)] reasons over populations of entities to render the grounded theory computationally tractable. These methods generally do not support the concept of (continuous) soft semantics through the use of semantic embedding spaces, as DASL does, although DeepProbLog Manhaeve et al. (2018) uses semantic embeddings and neural predicate representations within Bayesian Prolog rather than full FOL.

## 3 APPROACH

In this section we describe our approach to integrate data with relevant expert knowledge. Consider the task, depicted in Figure 1, of predicting the relationship between bounding boxes containing a subject and an object. In addition to learning from labeled training samples, we want to incorporate the commonsense knowledge that if the predicted relationship is "Riding" then the subject must be able to ride, the object must be ridable, and the subject must be above the object. Incorporating such knowledge results in a more robust model that uses high-level semantics to improve generalization and learn from a small number of examples. DASL achieves integration of the continuous representations in DNNs with the discrete representations typically used for knowledge representation by compiling a DNN from the knowledge assertions and grounding the vocabulary of the domain in component networks, enabling gradient-based learning of the model parameters.

We begin by providing the theoretic underpinning of DASL in first-order logic. Full FOL with equality, as defined for example by the use of Tarski models, is more expressive than SQL, SPARQL, Datalog, Prolog, and OWL, all of which can be recognized as "fragments" of FOL. We demonstrate here that DASL captures the full power of FOL with equality (*i.e.*, that it reasons correctly over all assertions expressible in FOL when '=' is taken to be defined as equality rather than an interpretable relation) by proving that DASL models are equivalent to Tarski models.

### 3.1 DASL MODEL THEORY

We follow a standard treatment of first order logic, starting with a language $\mathfrak{L}$ consisting of variables, constants, function symbols, and relation symbols. We use standard connectives $\neg$ ('not'), $\wedge$ ('and'), $\vee$ ('or'), $\rightarrow$ ('implies'), and quantifiers $\forall$ ('for all') and $\exists$ ('there exists'). For simplicity, we assume that no variable occurs bound twice, meaning that we would always need to rewrite $(\forall x)\varphi(x) \wedge (\exists x)\psi(x)$ as $(\forall x)\varphi(x) \wedge (\exists y)\psi(y)$. We treat $=$ ('equals') as a logical relation symbol. Terms are built up from variables, constants, and function symbols. Formulas are build up from terms, realtion symbols, and connectives. We restrict our attention to finite languages $\mathfrak{L}$ and only consider finite formulas over $\mathfrak{L}$.

Terms and formulas provide the syntax of first order logic. The semantics for classical logic is provided by boolean *Tarski models* [Chang & Keisler (1973), Weiss & D'Mello (1997)], a special case of infinite-valued Tarski models in which *truth values* range over the closed interval $\mathbb{T} = [0, 1]$ [Hájek (1998)]. A Tarski model $\mathfrak{A}$ with *domain* $A$ consists of a function $I$ which maps constants to elements of $A$, an $n$-ary function $I_f : A^n \rightarrow A$ for each $n$-ary function symbol $f$, and an $n$-ary function $I_R : A^n \rightarrow \mathbb{T}$ for each $n$-ary relation symbol $R$. We write $\mathfrak{A} = \langle A; I, I_{f_1}, \ldots, I_{f_m}, I_{R_1}, \ldots, I_{R_r}\rangle$ and $|\mathfrak{A}| = A$. A *sample function* maps variables to $A$; a model $\mathfrak{A}$ and set $\mathfrak{S}$ of sample functions together map terms to $A$ and formulas to $\mathbb{T}$, written $(\mathfrak{A}, \mathfrak{S})[s]$ and $(\mathfrak{A}, \mathfrak{S})[\varphi]$.

The logical symbols need a fixed interpretation in order to specify a full semantics. We define $(\mathfrak{A}, \mathfrak{S})[\neg\varphi] = 1 - (\mathfrak{A}, \mathfrak{S})[\varphi]$; $(\mathfrak{A}, \mathfrak{S})[\varphi \wedge \psi] = (\mathfrak{A}, \mathfrak{S})[\varphi] * (\mathfrak{A}, \mathfrak{S})[\psi]$; and $(\mathfrak{A}, \mathfrak{S})[(\forall x)\varphi] = \prod_{u \in A}(\mathfrak{A}, \mathfrak{S} \upharpoonright u/x)[\varphi]$ where $\mathfrak{S} \upharpoonright u/x = \{\mathcal{S} \mid \mathcal{S} \in \mathfrak{S}, \mathcal{S}(x) = u\}$, so we are interpreting universal statements as (possibly infinite) conjunctions over all possible instantiations of the bound variable. $\rightarrow$, $\vee$, and $\exists$ are defined in terms of $\neg$, $\wedge$, and $\forall$. An interpretation of '=' is said to *preserve equality* if for any terms $t$ and $s$, $(\mathfrak{A}, \mathfrak{S})[t = s] = 1$ if and only if $(\mathfrak{A}, \mathfrak{S})[t] = (\mathfrak{A}, \mathfrak{S})[s]$ for every $\mathfrak{A}$ and $\mathfrak{S}$. For now, we assume an equality-preserving interpretation, and we specify a particular interpretation below. We write $\mathfrak{S}^*(A)$ for the class of all sample functions over $A$ and $\mathfrak{A}[\varphi]$ for $(\mathfrak{A}, \mathfrak{S}^*(|\mathfrak{A}|))[\varphi]$. $(\mathfrak{A}, \mathfrak{S})[\varphi]$ is well-defined whenever every $\mathcal{S} \in \mathfrak{S}$ is defined over every (bound and free) variable of $\varphi$ and is consistent over every free variable of $\varphi$, and in particular $\mathfrak{A}[\varphi]$ is well-defined if and only if $\varphi$ is *closed* (meaning that every variable of $\varphi$ is bound by a quantifier).

**Proposition 1** *Let $\varphi$ be a closed formula. Every infinite-valued Tarski model $\mathfrak{A}$ such that $\mathfrak{A}[\varphi] = 1$ defines a classical boolean Tarski model of $\varphi$, and every boolean Tarski model of $\varphi$ defines an infinite-valued Tarski model mapping $\varphi$ to 1.*

This establishes that infinite-valued Tarski models fully capture the semantics of classical first order logic. While such models are not novel, their definitions vary significantly, and our use of sample functions instead of the usual variable interpretations is customized for our purposes below.

### 3.2 FINITARY COMPUTATION

DASL unifies machine learning and logical inference by generating infinite-valued Tarski models as described above which best fit both training data and logical formulas. A finite theory $\Gamma$ is represented in DASL by the conjunction of its formulas, so we can focus on operation over a single formula $\varphi$. An embedding architecture is specified for the interpretation function $I$, and a network architecture is specified for each $I_F$. The arithmetic semantic operators described above are applied as specified by the structure of $\varphi$, composing the interpretation networks into a single network, and this network is trained with a target output of 1.

We would like to prove that this technique completely captures full first order logic, but of course we know that some logical theories are only satisfied by infinite domains, and that all of our computation will be done over a finite number of finite samples. The constructed DASL network together with assigned parameter values specifies a unique model $\mathfrak{A}$, so any optimization technique that moves from one set of parameter assignments to another defines a sequence of models $\mathfrak{A}_0, \mathfrak{A}_1, \ldots$, each of which is evaluated over some finite set of samples, usually a "mini-batch" consisting of a relatively small subset of the domain. In order to relate this approach to infinitary models, we need to consider convergence in the limit. We begin by fixing a domain $A \subseteq \mathbb{R}^N$ for some $N$. If each $\mathfrak{A}_i$ is $\langle A; I_i, I_{1,i}, \ldots, I_{m,i} \rangle$ then we define $\lim_{i \to \infty} \mathfrak{A}_i = \langle A; \lim_{i \to \infty} I_i, \lim_{i \to \infty} I_{1,i}, \ldots, \lim_{i \to \infty} I_{m,i} \rangle$.

**Proposition 2** (Soundness) *Let $\mathfrak{A}_0, \ldots$ be a model sequence such that $\lim_{i \to \infty} \mathfrak{A}_i = \mathfrak{A}^*$ and $\mathfrak{S} = \bigcup_{i=0}^{\infty} \mathfrak{S}_i$ be a sample set such that every $\mathcal{S} \in \mathfrak{S}$ occurs in infinitely many of the $\mathfrak{S}_i$. For any formula $\varphi$, if $\lim_{i \to \infty} (\mathfrak{A}_i, \mathfrak{S}_i)[\varphi] = 1$ then $(\mathfrak{A}^*, \mathfrak{S})[\varphi] = 1$.*

The above proposition assumes that all possible samples are continually revisited, but makes no other assumptions about sampling. It makes no assumptions about how we train models or whether training converges. Together with proposition 1, this shows that when DASL models do converge on finite sampling, they converge to a boolean Tarski model.

**Proposition 3** (Completeness) *For any boolean model $\mathfrak{A}$ and formula $\varphi$ such that $\mathfrak{A}[\varphi] = 1$, there is a family of finite sample sets $\mathfrak{S}_0, \ldots$ such that $\bigcup_{i=0}^{\infty} \mathfrak{S}_i = \mathfrak{S}^*(|\mathfrak{A}|)$ and $(\mathfrak{A}, \mathfrak{S}_i)[\varphi] = 1$ for all $i$.*

The above proposition is proved by including in each set the sample function which contains a witness for each existentially bound variable whenever one exists. Details are in section A.1.1. This tells us that for any satisfiable formula, we can find the model that satisfies it even with only finite sampling.

### 3.3 IMPLEMENTING DASL NETWORKS

We implement DASL in the popular deep learning library PyTorch [Paszke et al. (2019)]. Any model that can be implemented in PyTorch can be used for the interpretation functions. We use a PyTorch `dataloader` to generate the mini-batches $\mathfrak{S}_i$ on which each model $\mathfrak{A}_i$ is evaluated.

**Loss function.** A simple and intuitive loss would be $\mathcal{L}(\mathfrak{A}, \mathfrak{S}, \varphi) = 1 - (\mathfrak{A}, \mathfrak{S})[\varphi]$. This loss works as desired at 0 and 1, but if $(\mathfrak{A}, \mathcal{S})[\varphi] = t$ for every $\mathcal{S} \in \mathfrak{S}$ then $(\mathfrak{A}, \mathfrak{S})[(\forall x)\varphi] = t^{|\mathfrak{S}|}$ which has the disadvantage that loss depends on the sample size and that it gets close to 1 as the sample size increases which can cause rounding errors to force a 0 gradient. Both of these challenges are addressed by using a logit representation of the truth value and using cross-entropy as a loss function. For $t \in \mathbb{T}$ define $\text{logit}(t) = \ln \frac{t}{1-t}$ and $\sigma(x) = \frac{1}{1+e^{-x}}$. Then $\sigma(\text{logit}(t)) = t$ and $\text{logit}(\sigma(x)) = x$. The cross entropy between the distributions $(t, 1-t)$ and $(1, 0)$ is $-\ln(t)$, so when $t = (\mathfrak{A}, \mathfrak{S})[\varphi]$ we define $\mathcal{L}(\mathfrak{A}, \mathfrak{S}, \varphi) = -\ln(t) = \ln(1 + e^{-\text{logit}(t)})$. We reason directly in logit space when computing the loss, deriving formulas from the above definitions (see subsection A.2 for more details).

**Equality.** DASL functions can include standard regression tasks $f(x) = y$ for real-valued $y$. This requires a differentiable interpretation that preserves equality. For (scalar-valued) terms $u$ and

$v$, we model $d = (\mathfrak{A}, \mathfrak{S})[u] - (\mathfrak{A}, \mathfrak{S})[v]$ by a normal distribution with mean 0 and variance $\varepsilon_1^2$ when equality holds and by a normal distribution with mean $\mu$ and variance $\varepsilon_2^2$ otherwise. We define the logit as the log ratio of the resulting densities for $x = |d|$: $\text{logit}((\mathfrak{A}, \mathfrak{S})[u = v]) = \ln \frac{2\varepsilon_2^2}{\varepsilon_1} + \frac{x^2}{2\varepsilon_1} - \ln(e^{-(x-\mu)^2/2\varepsilon_2^2} + e^{-(x+\mu)^2/2\varepsilon_2^2})$. For vector-valued $u$ and $v$, we use the same distribution with $x = \|u - v\|$. This interpretation approaches preserving equality as $\varepsilon_1 \to 0$.

**Extending the logical language.** We implement many-sorted logic for convenience and efficiency. Sorts are subsets of the domain. Each quantifier ranges over a specified subset. Quantifiers can map variables to elements of tuples from enumerated sets, so a standard machine learning problem can be written as $(\forall(x, y) : D)f(x) = y$, where $D$ lists the $(x, y)$ training pairs.

In addition to learned models, arbitrary deterministic functions can be coded in PyTorch. This allows us to fix logical interpretations of $<, >, +, -, \times$, etc. We allow propositional constants (formulas of the language which are always interpreted either as 0 or as 1) and we allow $\wedge$ and $\vee$ to apply to arbitrary sequences of formulas. The connectives also generalize to work component-wise on tensors and support broadcasting as is familiar for tensor operators in PyTorch. For example, if $X$ is a matrix and $y$ is a vector, both of logits, then $X \wedge y = Z$, where $Z_{ij} = X_{ij} \wedge y_i$.

Finally, we provide an explicit operator $softselect(\Gamma, i)$ (denoted as $\pi_i(\Gamma)$) which outputs the logit value for the $i^{\text{th}}$ formula of $\Gamma$ after application of the logit version of the softmax operator. This allows us to directly specify standard architectures for multi-class classification problems and to allow rules to operate on the classifier output within the model. Because $i$ is an integer argument, we can quantify over it, effectively quantifying over a fixed finite list of predicates, providing syntactic convenience without violating the constraints of first order logic.

## 4 EXPERIMENTS

We evaluate DASL on two computer vision problems in conditions of data scarcity. The first task is a toy problem based on MNIST digit classification, where knowledge is provided as an arithmetic relation satisfied by unlabeled triplets of digit images that are arranged to satisfy that relation (subsection 4.1). We then focus on the problem of detecting visual relationships between object pairs and use commonsense knowledge about the plausible arguments of the relationship (subsection 4.2).

### 4.1 TOY EXAMPLE ON MNIST

**Problem statement:** We use a toy example to demonstrate DASL's ability to train a NN from a few labeled samples and large number of unlabeled samples satisfying a rule. We denote a grayscale input image of a MNIST digit [LeCun et al. (1998)] as $X$ and its label (if provided) as $y(X) \in \mathbb{Z}_{10}$, where $\mathbb{Z}_{10} = \{0, 1, ..., 9\}$. The task is to learn a NN $\text{digit}(X)$ to predict the digit in a test image.

We split the training data ($50K$ images) into two disjoint sets: **Labeled**, containing a small number $N_{tr}$ of labeled examples per digit class, and **Unlabeled**, used to generate the set **Triples** containing triplets of images $(X_1, X_2, X_3)$ satisfying the rule $y(X_1) + y(X_2) = y(X_3) \bmod 10$. **Triples** contains only unlabeled images that together satisfy this relationship. We wish to learn the classifier by using **Labeled** and **Triples**, and thus the challenge is to compensate for the small size of **Labeled** by leveraging the prior knowledge about how the unlabeled images in **Triples** are related. We formulate this problem within DASL by using its $softselect$ operator $\pi_i$ that, applied to the NN output $\text{digit}(X)$, returns the normalized score for the $i^{th}$ class. This rule is written as:

$$(\forall(X_1, X_2, X_3) : \textbf{Triples})(\forall y_1 : \mathbb{Z}_{10})(\forall y_2 : \mathbb{Z}_{10})$$
$$[(\pi_{y_1}(\text{digit}(X_1)) \wedge \pi_{y_2}(\text{digit}(X_2)))$$
$$\to \pi_{(y_1 + y_2) \bmod 10}(\text{digit}(X_3))]$$

We quantify over the triplets from **Triples** and all possible pairs of digits from $\mathbb{Z}_{10}$. We use this theory to augment the theory corresponding to the labeled training examples $(\forall(X) : \textbf{Labeled})(\pi_{y(X)}(\text{digit}(X)))$. The model is required to correctly infer the (unknown) labels of the triplet members and then use them for indirect supervision. We evaluate the model using the average accuracy on the test set ($10K$ images). For $\text{digit}(X)$, we used a two-layer perceptron with $512$ hidden units and a sigmoid non-linearity. We performed experiments in data scarce settings

with $N_{tr} = 2, 5, 10$, and 20, and report mean performance with standard deviation across 5 random training subsets as shown in Figure 2. We use an equal number of examples per-class for constructing the triplets. We use a curriculum based training strategy to prevent the model from collapsing to a degenerate solution, especially for lower values of $N_{tr}$ (see subsection A.4 for details). We report performance after $30K$ training iterations. A test image is classified into the maximum scoring class.

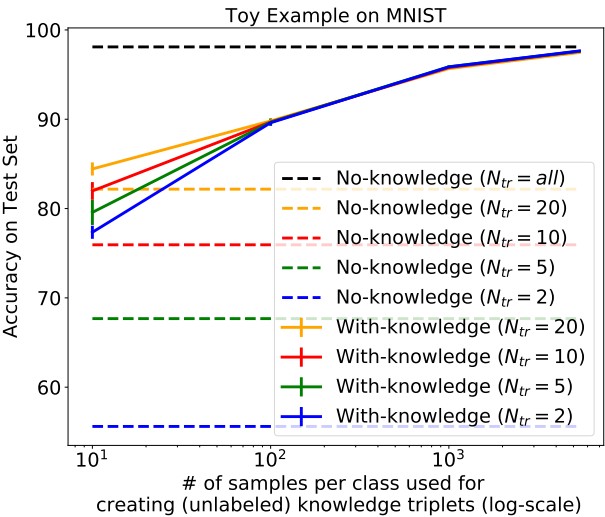

Figure 2: Figure showing the results for the MNIST toy example with a plot of accuracy of digit classification versus number of samples per class used for creating the unlabeled knowledge triplets. The labels *With-knowledge* and *No-knowledge* denote whether the training included the knowledge-augmented unlabeled triplets satisfying the given modular arithmetic (see subsection 4.1). $N_{tr}$ refers to the number of labeled training examples per class (*all* refers to the entire training set). Best seen in color.

**Results:** Figure 2 shows a plot of digit classification accuracy versus the number of samples per class used for creating the triplets. We observe that the NN trained with both knowledge and data (*With-knowledge*) outperforms its counterpart trained with only labeled samples (*No-knowledge*). The improvement is particularly notable when training with smaller labeled training sets; *e.g.*, for $N_{tr} = 2$, using all the knowledge raises performance from $53.3 \pm 1.01$ to $97.7 \pm 0.00$. We also note that the performance of the *With-knowledge* model improves as the number of triplets increases and converges to similar values for different values of $N_{tr}$, indicating that the knowledge renders extra labels largely superfluous. The mean performance is $97.6 \pm 0.00$, which is competitive with the performance of a model trained with all $50K$ labeled examples in MNIST ($98.1$ for $N_{tr} = all$). Results on a related problem were also shown in [Manhaeve et al. (2018)].

## 4.2 VISUAL RELATIONSHIP DETECTION

**Problem Statement:** We use the Visual Relationship Detection (VRD) benchmark [Lu et al. (2016)] to evaluate the **Predicate Detection Task:** Given an image and a set of bounding boxes with object category labels, predict the predicates that describe the relationships within object pairs. The VRD dataset contains 5000 images spanning 37993 relationships covering 100 object classes and 70 predicate classes. We use splits provided by the authors that contain 4000 train and 1000 test images. The dataset also provides a *zero-shot* test subset of 1877 relationships built from the same classes as the training data but containing novel combinations of predicate classes with object class pairs.

**Baseline model:** We begin with a NN $\mathrm{vrd}(\boldsymbol{I}, s, o)$ that outputs raw scores for predicate classes, where $\boldsymbol{I}$ is the input RGB image and $s$ and $o$ are the indices of the subject and object classes respectively. We implement two variants of $\mathrm{vrd}$ similar to that proposed in [Liang et al. (2018)]. The first variant, referred to as *VGG*, extracts visual features from the last layer of a pre-trained VGG-16 network from the bounding box of the subject, the object, and their union. These features are projected into a 256 dimensional space by using a projection layer $P$ (made of a fully-connected (FC) layer and a ReLU non-linearity) and then fused by concatenation. The features are passed through another $P$ layer

followed by a FC layer to predict the class-wise scores. The second variant, referred to as *VGG-SS*, additionally incorporates the word-embedding features of the subject and the object (300 dimensional Glove features [Pennington et al. (2014))] along with the normalized relative spatial coordinates (see subsection A.5 for details). These features are first projected using additional $P$ layers and then concatenated with visual features, as done for VGG, prior to predicting the class-scores.

We express knowledge as vectors of formulas as discussed in subsection 3.3. We begin by defining $\mathrm{CanRide}$ as a constant vector of truth values for all objects which is *True* at indices of objects which can ride and *False* elsewhere. $\mathrm{CanRide}(s)$ selects its $s^{\mathrm{th}}$ element. Similarly, we define $\mathrm{Ridable}$ as a vector which is *True* at exactly the indices of objects that can be ridden. Finally we define a one-hot vector of truth values $\boldsymbol{h}_{Cls} \in \mathbb{R}^{70}$, which is *True* at the index of the predicate class "Cls" and *False* elsewhere. The theory that asserts that vrd should output the class labels assigned in the training data and that the "Riding" predicate should only apply when the subject can ride and the object can be ridden is written as:

$$(\forall (\boldsymbol{I}, s, o, y) : \mathbf{D}_{\mathrm{vrd}})[\pi_y(\mathrm{vrd}(\boldsymbol{I}, s, o)$$
$$\wedge (\boldsymbol{h}_{\mathrm{Riding}} \to \mathrm{CanRide}(s)$$
$$\wedge \mathrm{Ridable}(o)))]$$

where $y$ is the given training label and $\mathbf{D}_{\mathrm{vrd}}$ is the training dataset. This rule reduces the learning burden of the classifier for "Riding" class by allowing feedback only when $\mathrm{CanRide}(s)$ is *True*. We introduce a few more rules by adding them in conjunction with the above rules (see subsection A.6).

**Evaluation:** We follow [Yu et al. (2017)], reporting Recall@N (R@N), the recall of the top-N prediction scores in an image where we take into account all 70 predictions per object pair. This strategy is different from [Lu et al. (2016)], which only considers the top prediction for each object pair penalizing cases where multiple predicates apply equally well but were omitted by the annotators.

| Method | R@50 | | R@100 | |
|---|---|---|---|---|
| | Standard | Zero-Shot | Standard | Zero-Shot |
| | 1% Data | | | |
| VGG (baseline) | $60.8 \pm 6.7$ | $40.7 \pm 5.8$ | $75.4 \pm 7.8$ | $59.4 \pm 8.1$ |
| + Knowledge | $68.5 \pm 1.8^{**}$ | $49.5 \pm 1.5^{**}$ | $83.1 \pm 1.6^{**}$ | $70.1 \pm 2.4^{**}$ |
| VGG-SS (baseline) | $67.9 \pm 8.5$ | $47.6 \pm 8.5$ | $80.3 \pm 7.6$ | $65.6 \pm 9.2$ |
| + Knowledge | $74.0 \pm 0.7^{*}$ | $54.4 \pm 1.4^{*}$ | $85.9 \pm 0.5^{*}$ | $73.4 \pm 1.2^{*}$ |
| | 5% Data | | | |
| VGG (baseline) | $70.3 \pm 0.5$ | $48.4 \pm 1.0$ | $83.5 \pm 0.4$ | $68.3 \pm 0.9$ |
| + Knowledge | $73.8 \pm 0.5^{**}$ | $53.4 \pm 0.9^{**}$ | $86.4 \pm 0.4^{**}$ | $73.7 \pm 1.1^{**}$ |
| VGG-SS (baseline) | $79.6 \pm 0.4$ | $58.1 \pm 1.2$ | $89.6 \pm 0.3$ | $77.1 \pm 1.1$ |
| + Knowledge | $79.9 \pm 0.4$ | $59.6 \pm 0.9^{**}$ | $89.7 \pm 0.3$ | $78.5 \pm 0.8^{**}$ |

Table 1: Performance on the predicate detection task [Lu et al. (2016)]. We report Recall@N averaged (with standard deviation) across 10 random subsets. We report the statistical significance between "baseline" and corresponding knowledge augmented model ("+ Knowledge") (p-value $< 0.01$ as $**$ and p-value $< 0.05$ as $*$).

**Results:** Table 1 shows the results on the VRD dataset when training with knowledge (+ *Knowledge*) and without knowledge (*baseline*). We observe consistent improvements across all cases with augmentation of knowledge. The improvements are higher for the 1% data (+7.7% for R@100 for Standard) than the 5% data (+2.9% for R@100 for Standard) showing that knowledge has more benefits in lower data regimes. We made similar observation for the MNIST toy example. The improvements are generally higher for the zero-shot setting (+10.7% for R@100 in the 1% case) since this setting is inherently data starved. We also note that the improvements are comparatively smaller for the VGG-SS network since semantic and spatial information are being explicitly injected as features into the model. We provide class-wise performance for a few predicates in subsection A.8. We observe consistent improvement for predicates such as "Wear" and "Above" that were specified in the rules.

### 4.3 COMPARISON WITH RELATED NEURO-SYMBOLIC METHODS

We empirically verified the advantage of the logit representation over two related methods [Serafini & Garcez (2016); Demeester et al. (2016)], both of which use product-t norms but represent the

final truth values using either probabilities or log probabilities. We evaluated these methods on the task of learning a digit classifier through two variants of logical conjunctions that make assertions about images and their ground-truth labels. Both of these these techniques fail catastrophically when the number of conjuncts (see Figure 3) is increased. In contrast, DASL did not see any loss in performance, highlighting its better numerical stability. We refer readers to subsection A.7 for details.

We also compare our model with a recent neuro-symbolic method [Xie et al. (2019)] that uses graph neural networks with propositional formulas to semantically regularize output embeddings for VRD. We observe clear improvements in the top-5 accuracy metric used in their work under similar training conditions. With $5\%$ data, our model ("VGG-SS + Knowledge") achieves $77.7\%$ accuracy versus $72.4\%$ of [Xie et al. (2019)]. The improvements are higher for $1\%$ data– $71.5\%$ accuracy of ours versus $55.5\%$ of [Xie et al. (2019)]– highlighting the advantages of DASL with a small amount of training data. We refer readers to subsection A.8 for details.

## 5 CONCLUSION

We introduced **Deep Adaptive Semantic Logic (DASL)** to unify machine reasoning and machine learning. DASL is fully general, encompassing all of first order logic and arbitrary deep learning architectures. It improves deep learning by supplementing training data with declarative knowledge expressed in first order logic. The vocabulary of the domain is realized as a collection of neural networks. DASL composes these networks into a single DNN and applies backpropagation to satisfy both data and knowledge. We provided a formal grounding that demonstrates the correctness and full generality of DASL for the representation of declarative knowledge in first order logic, including correctness of mini-batch sampling for arbitrary domains. This gives us the freedom to apply DASL to new domains without requiring new correctness analysis.

We demonstrated a 1000-fold reduction in data requirements on the artificially constructed MNIST digit classification task by using declarative knowledge in the form of arithmetic relations satisfied by unlabeled image triplets. We then demonstrated the application of commonsense knowledge to visual relationship detection, improving recall from $59.4\%$ to $70.1\%$. Here, the commonsense knowledge enabled us to better focus the model on cases not covered by knowledge.

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

# A APPENDIX

## A.1 FORMAL SEMANTICS FOR DASL

### A.1.1 SAMPLE FUNCTIONS

We assume the definitions provided in Section 3 and make explicit some of the details of sample functions. We fix the set $V = \{x_1, \ldots\}$ to be the set of variables, independent of any language $\mathfrak{L}$. Fix a language $\mathfrak{L}$ with constants $c_1, \ldots$, function symbols $f_1, \ldots$ with arities $n_{f_1}, \ldots$, and relation symbols $R_1, \ldots$ with arities $n_{R_1}, \ldots$. Let $\mathfrak{A}$ be the model $\langle A; I, I_{f_1}, \ldots, I_{R_1}, \ldots \rangle$.

A *sample function for* $\mathfrak{A}$ is a function $\mathcal{S} : V \to A$. For $V_0 \subseteq V$ we say that sample functions $\mathcal{S}_1$ and $\mathcal{S}_2$ are *consistent over* $V_0$ if $\mathcal{S}_1(v) = \mathcal{S}_2(v)$ for every $v \in V_0$. We say that a set $\mathfrak{S}$ of sample functions is *consistent over* $V_0$ if every pair of sample functions in $\mathfrak{S}$ is consistent over $V_0$. For any term or formula $\Phi$ we say $\mathfrak{S}$ is *consistent for* $\Phi$ if $\mathfrak{S}$ is consistent over the set of free variables of $\Phi$.

Let $s_1, \ldots$ be any terms of $\mathfrak{L}$ and $\mathfrak{S}$ be an arbitrary set of sample functions for $\mathfrak{A}$. We define for every $i$:

$$
\begin{aligned}
(\mathfrak{A}, \mathfrak{S})[x_i] &= \mathcal{S}(x_i) \text{ for } \mathcal{S} \in \mathfrak{S} \text{ if } \mathfrak{S} \text{ consistent for } \{x_i\} \\
(\mathfrak{A}, \mathfrak{S})[a] &= I(a) \\
(\mathfrak{A}, \mathfrak{S})[f_i(s_1, \ldots, s_{n_{f_i}})] &= \mathcal{I}_{f_i}((\mathfrak{A}, \mathfrak{S})[s_1], \ldots, (\mathfrak{A}, \mathfrak{S})[s_{n_{f_i}}])
\end{aligned}
$$

Under this definition, the pair $(\mathfrak{A}, \mathfrak{S})$ maps every term of $\mathfrak{L}$ to a member of $A$ (recalling from section 3.1 that $I$ maps constants to $A$) whenever $\mathfrak{S}$ is consistent for the term. The map is undefined when $\mathfrak{S}$ is not consistent for the term. It can be seen by recursion on the definition that whenever $\mathfrak{S}$ is consistent for the term on the left-hand side of any of the above equations, it is also consistent for all terms on that equation's right-hand side. It can also be seen that for any term $s$, $(\mathfrak{A}, \{\mathcal{S}\})[s] = (\mathfrak{A}, \mathfrak{S})[s]$ for every $\mathcal{S} \in \mathfrak{S}$ whenever $\mathfrak{S}$ is consistent for $s$.

For brevity of presentation, we define:

$$
\begin{aligned}
\varphi \vee \psi &\equiv \neg(\neg\varphi \wedge \neg\psi) \\
\varphi \to \psi &\equiv \neg(\varphi \wedge \neg\psi) \\
(\exists x)\varphi &\equiv \neg(\forall x)\neg\varphi
\end{aligned}
$$

We extend $(\mathfrak{A}, \mathfrak{S})$ to also map formulas to truth values. Since $\mathfrak{A}$ is boolean, the interpretation of $=$ needs to always evaluate to 0 or 1. There is only one such interpretation that preserves equality. Define:

$$
\begin{aligned}
(\mathfrak{A}, \mathfrak{S})[R_i(s_1, \ldots, s_{n_{R_i}})] &= \mathcal{I}_{R_i}((\mathfrak{A}, \mathfrak{S})[s_1], \ldots, (\mathfrak{A}, \mathfrak{S})[s_{n_{R_i}}]) \\
(\mathfrak{A}, \mathfrak{S})[s_1 = s_2] &= \begin{cases} 1 & \text{if } (\mathfrak{A}, \mathfrak{S})[s_1] = (\mathfrak{A}, \mathfrak{S})[s_2] \\ 0 & \text{otherwise} \end{cases} \\
(\mathfrak{A}, \mathfrak{S})[\neg\varphi] &= 1 - ((\mathfrak{A}, \mathfrak{S})[\varphi]) \\
(\mathfrak{A}, \mathfrak{S})[\varphi \wedge \psi] &= (\mathfrak{A}, \mathfrak{S})[\varphi] \cdot (\mathfrak{A}, \mathfrak{S})[\psi] \\
(\mathfrak{A}, \mathfrak{S})[(\forall x)\varphi] &= \prod_{u \in A} (\mathfrak{A}, \mathfrak{S} \upharpoonright u/x)[\varphi]
\end{aligned}
$$

The application of $(\mathfrak{A}, \mathfrak{S})$ to formulas depends on its application to terms, and therefore is still only well-defined if $\mathfrak{S}$ is consistent for the formula to which it is applied. As above, we can see by recursion that whenever $\mathfrak{S}$ is consistent for the left-hand side of any equation other than the last (the $\forall$ equation) it is consistent all formulas on the right-hand side of that equation. For $\forall$ we note that when $\mathfrak{S}$ is consistent for $(\forall x)\varphi$, then it is consistent for all free variables of $\varphi$ other than $x$. $\mathfrak{S}' = \mathfrak{S} \upharpoonright u/x$ is a subset of $\mathfrak{S}$, so it must also be consistent for all free variables of $\varphi$ other than $x$. By definition, for every $\mathcal{S} \in \mathfrak{S}'$, $\mathcal{S}(x) = u$, so $\mathfrak{S}'$ is consistent for $x$ as well and therefore is consistent for $\varphi$. Thus, the $\forall$ clause of the above definition decomposes the set $\mathfrak{S}$ of sample functions into a class of sets of sample functions, each consistent for $\varphi$, and then conjoins the results of applying each to $\varphi$. This particular technique is convenient because it allows us to specify sampling algorithms formally for implementation.

The intent of the interpretation of $\forall$ above is that we are applying all of the sample functions in $\mathfrak{S}$, but in general there could be some $u \in A$ for which $\mathfrak{S}$ does not contain any sample function that maps $x$ to $u$, with the result that some of the terms on the right hand side become $(\mathfrak{A}, \{\})[\varphi]$, which is undefined. In principle this could be resolved by taking the product over $\{u \mid \mathcal{S}(x) = u$ for some $\mathcal{S} \in \mathfrak{S}\}$, but instead we define $(\mathfrak{A}, \{\})[\varphi] = 1$ for all $\varphi$ just for notational convenience.

### A.1.2 FINITE SAMPLES

We will describe construction of finite sample sets that allow for complete search of the infinite model domain, proving proposition 3 from Section 3 in the main text. For an arbitrary set $\mathfrak{S}$ of sample functions, if $\mathcal{F} = \{\mathfrak{S}_i \mid i \in I\}$ is a family of finite sets and $\bigcup_{i \in I} \mathfrak{S}_i = \mathfrak{S}$ then $\mathcal{F}$ is called a *finite decomposition* of $\mathfrak{S}$. Given a model $\mathfrak{A}$ and formula $\varphi$, we say that $\mathcal{F}$ *preserves truth for $\mathfrak{A}$ and $\varphi$* if $(\mathfrak{A}, \mathfrak{S}_i)[\varphi] = (\mathfrak{A}, \mathfrak{S})[\varphi]$ for every $i$. If $\mathfrak{S}_i$ is consistent for a set $V$ of variables for every $i$ then we say that $\mathcal{F}$ *is consistent for $V$*.

We prove a generalization of proposition 3:

**Proposition 4** *Let $\mathfrak{A}$ be a boolean model, $A = |\mathfrak{A}|$, $\varphi$ be a formula, and $\mathfrak{S} = \{\mathcal{S}_i \mid i \in \mathcal{I}\}$ be a set of sample functions consistent for $\varphi$ for some infinite index set $\mathcal{I}$. Assume $|\mathfrak{S}| = |A|$. Then there is a finite decomposition $\{\mathfrak{S}_i \mid i \in \mathcal{I}\}$ of $\mathfrak{S}$ that preserves truth for $\varphi$ and is consistent for $\varphi$.*

Note that $|\mathfrak{S}^*(A)| = |A|$, so the assumption on the size of the sample set is a natural one and is met by proposition 3. To prove this claim we first note that whenever $\mathfrak{S}_0 \subseteq \mathfrak{S}$ and $\mathfrak{S}$ is consistent for $\varphi$ then $\mathfrak{S}_0$ is consistent for $\varphi$, so every finite decomposition of $\mathfrak{S}$ will be consistent for $\varphi$. We proceed by induction on the structure of $\varphi$.

**Suppose $\varphi$ is $R(s_1, \ldots, s_n)$ or $s_1 = s_2$.** Choose any $\mathcal{S} \in \mathfrak{S}$.

$$
\begin{aligned}
(\mathfrak{A}, \{\mathcal{S}\})[R(s_1, \ldots, s_n)] &= I_R((\mathfrak{A}, \{\mathcal{S}\})[s_1], \ldots, (\mathfrak{A}, \{\mathcal{S}\})[s_n]) \\
&= I_R((\mathfrak{A}, \mathfrak{S})[s_1], \ldots, (\mathfrak{A}, \mathfrak{S})[s_n]) \\
&= (\mathfrak{A}, \mathfrak{S})[R(s_1, \ldots, s_n)]
\end{aligned}
$$

Similarly, $(\mathfrak{A}, \{\mathcal{S}\})[s_1 = s_2] = 1$ if and only if $(\mathfrak{A}, \{\mathcal{S}\})[s_1] = (\mathfrak{A}, \{\mathcal{S}\})[s_2]$, which holds if and only if $(\mathfrak{A}, \mathfrak{S})[s_1] = (\mathfrak{A}, \mathfrak{S})[s_2]$. Thus in both cases $\{\{\mathcal{S}\} \mid \mathcal{S} \in \mathfrak{S}\}$ is a finite decomposition of $\mathfrak{S}$ that preserves truth for $\varphi$.

**Suppose $\varphi$ is $\neg\psi$.** Let $t = (\mathfrak{A}, \mathfrak{S})[\neg\psi]$, so $(\mathfrak{A}, \mathfrak{S})[\psi] = 1 - t$. By the induction hypothesis let $\mathcal{F} = \{\mathfrak{S}_i \mid i \in \mathcal{I}\}$ be a truth-preserving finite decomposition of $\mathfrak{S}$ for $\psi$.

$$
\begin{aligned}
(\mathfrak{A}, \mathfrak{S}_i)[\neg\psi] &= 1 - (\mathfrak{A}, \mathfrak{S}_i)[\psi] \\
&= 1 - (\mathfrak{A}, \mathfrak{S})[\psi] \\
&= (\mathfrak{A}, \mathfrak{S})[\neg\psi]
\end{aligned}
$$

Thus $\mathcal{F}$ also preserves truth for $\neg\psi$.

**Suppose $\varphi$ is $\psi_1 \wedge \psi_2$.** By the induction hypothesis, let $\{\mathfrak{S}_i^1 \mid i \in \mathcal{I}\}$ and $\{\mathfrak{S}_i^2 \mid i \in \mathcal{I}\}$ be finite decompositions of $\mathfrak{S}$ that preserve truth for $\psi_1$ and $\psi_2$, respectively. For each $i, j \in \mathcal{I}$ define:

$$
\mathfrak{S}_{i,j} = \{\mathcal{S}_1 \circ \mathcal{S}_2 \mid \mathcal{S}_1 \in \mathfrak{S}_i^1, \mathcal{S}_2 \in \mathfrak{S}_j^2\}
$$

where we define

$$
(\mathcal{S}_1 \circ \mathcal{S}_2)(x) = \begin{cases} \mathcal{S}_2(x) & \text{if } x \text{ is bound in } \psi_2 \\ \mathcal{S}_1(x) & \text{otherwise} \end{cases}
$$

Recall that all bound variables have been renamed to be distinct from each other and from all free variables. For each $i$ and $j$, $\mathfrak{S}_{i,j}$ and $\mathfrak{S}_i^1$ are identical over all free and bound variables of $\psi_1$, since none of these is bound in $\psi_2$. Thus $(\mathfrak{A}, \mathfrak{S}_{i,j})[\psi_1] = (\mathfrak{A}, \mathfrak{S}_i^1)[\psi_1] = (\mathfrak{A}, \mathfrak{S})[\psi_1]$. Similarly, $\mathfrak{S}_{i,j}$ is identical to $\mathfrak{S}_j^2$ over all free and bound variables of $\psi_2$, so $(\mathfrak{A}, \mathfrak{S}_{i,j})[\psi_2] = (\mathfrak{A}, \mathfrak{S}_j^2)[\psi_2] = (\mathfrak{A}, \mathfrak{S})[\psi_2]$. Thus

$$
\begin{aligned}
(\mathfrak{A}, \mathfrak{S}_{i,j})[\psi_1 \wedge \psi_2] &= (\mathfrak{A}, \mathfrak{S}_{i,j})[\psi_1] \cdot (\mathfrak{A}, \mathfrak{S}_{i,j})[\psi_2] \\
&= (\mathfrak{A}, \mathfrak{S})[\psi_1] \cdot (\mathfrak{A}, \mathfrak{S})[\psi_2] \\
&= (\mathfrak{A}, \mathfrak{S})[\psi_1 \wedge \psi_2]
\end{aligned}
$$

To see that $\{\mathfrak{S}_{i,j} \mid i, j \in \mathcal{I}\}$ is a finite decomposition of $\mathfrak{S}$, let arbitrary $\mathcal{S} \in \mathfrak{S}$ be given. Choose $i$ and $j$ such that $\mathcal{S} \in \mathfrak{S}_i^1$ and $\mathcal{S} \in \mathfrak{S}_j^2$. Then $\mathfrak{S}_{i,j}$ contains $\mathcal{S} \circ \mathcal{S}$, which is $\mathcal{S}$. Re-indexing the family requires mapping $\mathcal{I}^2$ to $\mathcal{I}$, which is non-problematic since $\mathcal{I}$ is infinite.

**Suppose $\varphi$ is $(\forall x)\psi$.** Let $t = (\mathfrak{A}, \mathfrak{S})[(\forall x)\psi]$ and $t_u = (\mathfrak{A}, \mathfrak{S} \upharpoonright u/x)[\psi]$ for each $u \in A$. By the induction hypothesis, for each $u \in A$ let $\{\mathfrak{S}_i^u \mid i \in \mathcal{I}\}$ be a finite decomposition of $\mathfrak{S} \upharpoonright u/x$ that preserves truth for $\mathfrak{A}$ and $\psi$. Note that $\mathfrak{S}_i^v \upharpoonright u/x = \mathfrak{S}_i^v$ if $u = v$ and is $\{\}$ otherwise, and recall that $(\mathfrak{A}, \{\})[\chi] = 1$ for every formula $\chi$ by definition, so $\prod_{u \in A}(\mathfrak{A}, \mathfrak{S}_i^v \upharpoonright u/x)[\chi] = (\mathfrak{A}, \mathfrak{S}_i^v)[\chi]$.

If $t = 1$ then $t_v = 1$ for every $v$, so:

$$
\begin{aligned}
(\mathfrak{A}, \mathfrak{S}_i^v)[(\forall x)\psi] &= \prod_{u \in A}(\mathfrak{A}, \mathfrak{S}_i^v \upharpoonright u/x)[\psi] \\
&= (\mathfrak{A}, \mathfrak{S}_i^v)[\psi] \\
&= (\mathfrak{A}, \mathfrak{S} \upharpoonright v/x)[\psi] \\
&= 1
\end{aligned}
$$

Thus $\mathcal{F} = \{\mathfrak{S}_i^v \mid i \in \mathcal{I}, v \in A\}$ is a finite decomposition of $\mathfrak{S}$ that preserves truth for $\varphi$. Since $|A| \leq |\mathcal{I}|$, $|\mathcal{F}| \leq |\mathcal{I}|^2$, so $\mathcal{F}$ can be re-indexed in $\mathcal{I}$.

If $t = 0$, then since $\mathfrak{A}$ is boolean, choose $w$ such that $t_w = 0$. For $v \in A$ define $\mathfrak{S}_i^{v,w} = \mathfrak{S}_i^v \cup \mathfrak{S}_i^w$. As above, note that $\mathfrak{S}_i^{v,w} \upharpoonright u/x$ is $\mathfrak{S}_i^u$ if $u$ is $v$ or $w$ and is $\{\}$ otherwise.

$$
\begin{aligned}
(\mathfrak{A}, \mathfrak{S}_i^{v,w})[(\forall x)\psi] &= \prod_{u \in A}(\mathfrak{A}, \mathfrak{S}_i^{v,w} \upharpoonright u/x)[\psi] \\
&= (\mathfrak{A}, \mathfrak{S}_i^v)[\psi] \cdot (\mathfrak{A}, \mathfrak{S}_i^w)[\psi] \\
&= (\mathfrak{A}, \mathfrak{S} \upharpoonright v/x)[\psi] \cdot (\mathfrak{A}, \mathfrak{S} \upharpoonright w/x)[\psi] \\
&= 0
\end{aligned}
$$

Thus $\{\mathfrak{S}_i^{v,w} \mid i \in \mathcal{I}, v \in A\}$ is a finite decomposition of $\mathfrak{S}$ that preserves truth for $\varphi$.

This shows that proposition 4 holds for every case of $\varphi$, and by induction that it holds for all $\varphi$. Proposition 3 of Section 3 is a direct consequence.

## A.2    DASL Models as Neural Networks

For truth values $t_1$ and $t_2$ and corresponding logits $l_1$ and $l_2$, we define negation ($\neg$) and conjunction ($\wedge$) operators as:

$$\neg l_1 = \text{logit}(1 - t_1) = -l_1$$

$$l_1 \wedge l_2 = \text{logit}(t_1 t_2) = \ln \sigma(l_1) + \ln \sigma(l_2) - \ln(1 - \sigma(l_1)\sigma(l_2))$$

This formula for $\wedge$ is numerically unstable when $t_1 t_2$ gets close to 1. Whenever this occurs, we instead use the numerically stable approximation:

$$l_1 \wedge^* l_2 \approx -\ln(e^{-l_1} + e^{-l_2}).$$

We use PyTorch functions `logsigmoid` and `logsumexp` that provide efficient and numerically robust computations for terms arising in these equations.

Conjunction and universal quantification are naturally represented as products of truth values, but the product of a large number of positive terms all less than 1 gets arbitrarily close to 0, and so does its derivative, meaning that learning is slow or will stop altogether. Under the logit space equations above, however, conjunctions are sums, so increasing the number of conjuncts does not diminish the gradient. Two typical alternatives for $t_1 \wedge t_2$ in systems that operate directly on truth values are $\min(t_1, t_2)$ and $\max(0, t_1 + t_2 - 1)$ [Serafini & Garcez (2016)]. When many terms are conjoined, the former formula yields gradient information only for the minimal truth value, and the second yields no gradient information at all whenever $t_1 + t_2 < 1$, again restricting the ability of a system to learn.

### A.3 DETAILS REGARDING TRAINING FOR MNIST TOY EXAMPLE AND VISUAL RELATIONSHIP DETECTION

For both problems we trained the model with the Adam optimizer [Kingma & Ba (2014)] with a learning rate of $5 \times 10^{-5}$. The batch size was set to $64$ and $128$ for the MNIST toy problem and visual relationship detection respectively.

### A.4 CURRICULUM LEARNING FOR MNIST TOY EXAMPLE

In subsection 4.1 we trained a NN for digit classification on the MNIST dataset in conditions of data scarcity. We used a few labeled samples and a large number of unlabeled triplets satisfying some rules (modular arithmetic in our experiments). We used a curriculum based learning strategy to prevent the model from collapsing to a degenerate solution, especially for cases with an extremely small number of labeled samples (*e.g.* $2$ samples per class). In such cases the model tends to get trapped in a local minimum where the axiom corresponding to the unlabeled triplets can be satisfied by a solution with all digits being classified as $0$ since $0 + 0 = 0$. Within the curriculum, we begin the training with all the labeled examples and a small working set of the unlabeled triplets. We progressively expand the working set during training as the model becomes more confident on the unlabeled examples. The confidence score $p_c^t$ is computed using a low-pass filter:

$$p_c^t = (1 - \alpha) * p_c^{t-1} + \alpha * p_{max}$$

where $*$ is scalar multiplication, $t$ is the iteration index, $p_c^0 = 0$, $\alpha = 0.1$, and $p_{max}$ is the average probability of the highest scoring class on the first digit of the triplet. When $p_c^t > 0.9$, we increase the size of the working set of unlabeled triplets by a factor of $2$ until it reaches the maximum number of unlabeled triplets. When $p_c^t > 0.9$, we reset $p_c^t$ to $0$ to let the model fit well to the new working set before reaching the condition again. This curriculum ensures that the model is able to find a decent initialization using the labeled examples and then progressively improve using the unlabeled samples. The initial set of unlabeled triplets contained $10$ samples per class and the maximum number of triplets is bounded by the class with minimum number of samples. During the final curriculum step we remove all labeled data, allowing the model to train solely on the rules. This allows the model to trade off errors on the labeled data for better overall performance.

### A.5 NORMALIZED RELATIVE SPATIAL FEATURES FOR VISUAL RELATIONSHIP DETECTION

We provide the implementation details for the spatial features used in the visual relationship detection experiments in subsection 4.2. These features capture the relative spatial configuration of the subject and the object bounding boxes and were used to augment visual and semantic features for predicting the visual relationship (VGG-SS). We denote the coordinates of the object and subject bounding boxes as $(x_s, y_s, w_s, h_s)$ and $(x_o, y_o, w_o, h_o)$ respectively, where $(x, y)$ are the coordinates of the (box) center with width $w$ and height $h$. The relative normalized features are captured in an eight dimensional feature vector defined as $\left[ \frac{x_s - x_o}{w_o}, \frac{y_s - y_o}{h_o}, \frac{x_o - x_s}{w_s}, \frac{y_o - y_s}{h_s}, \log(\frac{w_s}{w_o}), \log(\frac{h_s}{h_o}), \log(\frac{w_o}{w_s}), \log(\frac{h_o}{h_s}) \right]$. These features were also used in the baseline model [Liang et al. (2018)].

### A.6 COMMONSENSE RULES FOR VISUAL RELATIONSHIP DETECTION

In addition to the rule for the "Riding" predicate described in subsection 4.2, we used rules for incorporating commonsense knowledge about a number of the other predicates in predicting visual relationships using DASL. These rules follow the same format as the rule for "Riding" and are outlined below:

1. "Wear" only applies when the subject is a *living entity* and the object is *wearable*.

2. "Sleep-On" only applies when the subject is a *living entity* and the object is *sleepable*.

3. "Eat" only applies when the object is *eatable*.

4. The predicates "Above", "Over", "Ride", "On-The-Top-Of", "Drive-on", "Park-On", "Stand-On", "Sit-On", and "Rest-On" apply only when the subject is spatially *above* the object. We defined *above* as a function that is *True* when $y_s \geq y_o$.

5. The predicates "Under", "Beneath", "Below", and "Sit-Under" apply only when the subject is spatially *below* the object. We defined *below* as a function that is *True* when $y_s \leq y_o$.

6. The predicate "On-The-Right-Of" applies only when the subject is spatially *right of* the object. We defined *right of* as a function that is *True* when $x_s \geq x_o$.

7. The predicate "On-The-Left-Of" applies only when the subject is spatially *left of* the object. We defined *left of* as a function that is *True* when $x_s \leq x_o$.

These rules cover facts related to both semantic and spatial commonsense knowledge. We incorporated these rules by conjoining them with the implication in the original theory presented in subsection 4.2.

$$
\begin{aligned}
(\forall(\boldsymbol{I}, s, o, y) : \mathbf{D}_{\mathrm{vrd}})[\pi_y(\mathrm{vrd}(\boldsymbol{I}, s, o) \\
\wedge (\boldsymbol{h}_{\mathrm{Riding}} \rightarrow \mathrm{CanRide}(s) \\
\wedge \mathrm{Ridable}(o)) \\
\wedge (\boldsymbol{h}_{\mathrm{Wear}} \rightarrow \mathrm{Living}(s) \\
\wedge \mathrm{Wearable}(o)) \ldots)]
\end{aligned}
$$

where $\boldsymbol{h}_{Cls} \in \mathbb{R}^{70}$ is a one-hot vector of truth values, which is *True* at the index of the predicate class "Cls" and *False* elsewhere. Living is a constant vector of truth values for all objects which is *True* at indices of objects which are living entities and *False* elsewhere. Similarly, Wearable is a constant vector, which is *True* at exactly the indices of objects which are wearable. We refer readers to subsection 4.2 for detailed explanation about the application of these rules.

We would like to note two approaches for building such knowledge bases. The first method is a manual method where a user can supply logical rules that are focused on classes where the model does not work well. The second method is to exploit knowledge bases such as Concept-Net, WordNet or meta-data of prior datasets such as Visual Genome [Krishna et al. (2017))]. For VRD, we started with Visual Genome to build our knowledge base and then kept a subset of rules which we found to be most helpful. While building these rules we found it helpful to group the predicates in a rule set. For example, predicates such as "Above" and "Over" were part of the "Above" rule set (number 4 in the list of rules) which verified whether the subject is above the object. We understand the general concern regarding the feasibility of constructing a knowledge base but in our view there is sufficient work in the literature showing how machine learning algorithms have exploited knowledge bases from pre-existing resources [Aditya et al. (2018); Moldovan & Rus (2001)].

### A.7 Empirical Comparison with Related Works

Here we compare DASL to two closely related neuro-symbolic based methods: Logic Tensor Networks (LTN) [Serafini & Garcez (2016)] and Rocktäschel et al. [Demeester et al. (2016)]. Both use product *t-norm* for the conjunction operator on truth values and cross-entropy for the loss function, as does DASL. These operations are carried out directly in LTN, in log space in Rocktäschel et al., and in logit space in DASL. Operating directly on the raw truth values suffers from vanishing gradients because the product of many values between 0 and 1 approaches 0. Operating on logs of truth values maintains precision near 0 but not near 1. The negation of conjunction has loss $\log(1 - \prod_i[A_i])$, which again has a vanishing gradient as the number of terms increases. DASL's logit transform maintains precision near both 0 and 1, enabling DASL to represent *full* FOL (in this case, including the negations of conjunctions) at scale. To verify these claims, we train the classification NN in subsection 4.1 on MNIST digits using two variants of logical conjunctions. The first variant trains the NN using a conjunction of multiple assertions about MNIST digits and their labels, *i.e.*, $(\forall(\boldsymbol{X}, y) : \mathbf{Labeled})(\pi_{y(\boldsymbol{X})}(\mathrm{digit}(\boldsymbol{X})))$. The second variant trains the NN using negation of conjunction for the same problem $\neg(\neg(\forall(\boldsymbol{X}, y) : \mathbf{Labeled})(\pi_{y(\boldsymbol{X})}(\mathrm{digit}(\boldsymbol{X}))))$. For a fair comparison we implement both the LTN and Rocktäschel et al. representations within the DASL's code by replacing DASL's logit space computations with truth value computations with probabilities and log-probabilities respectively for learning. To understand the sensitivity of different representations to the size of conjunction (batch-size in this case), we plot accuracies on the test-set against different batch-sizes in Figure 3. We see that the performance of LTN drops quickly as the number of assertions is increased for both variants since the raw representation fails to preserve

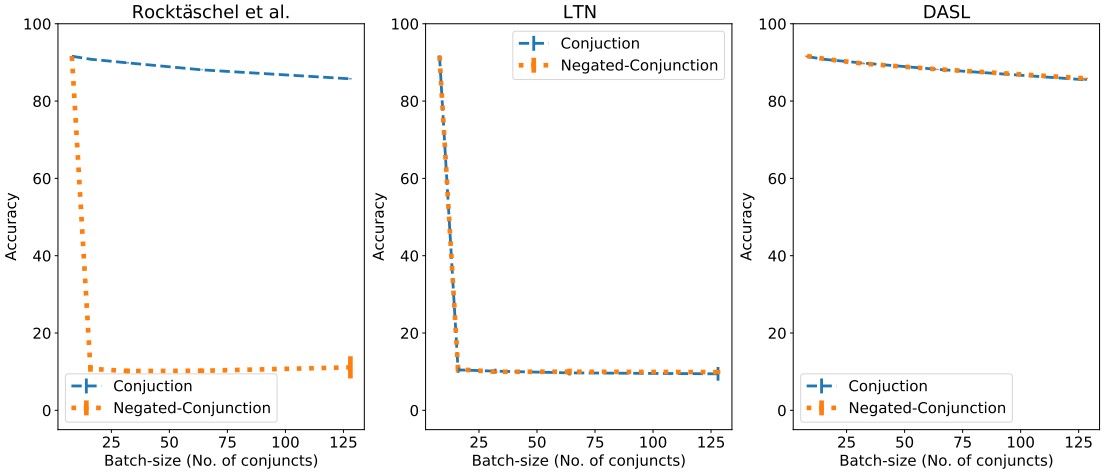

Figure 3: Comparison of DASL with LTN [Garcez et al. (2019)] and Rocktäschel et al. [Demeester et al. (2016)] for training a NN to classify MNIST digits with conjunctions and negated conjunctions over training samples. We observe that the performance of prior methods saturate when increasing the number of terms in conjunction. However, this is not the case with DASL due to its logit representation, which prevents vanishing gradients and thus enables improved learning.

gradients. Rocktäschel et al. performs similar to DASL for regular conjunction due to the use of log truth values. However, when training with negated conjunctions, the converged Rocktäschel et al. model fails catastrophically for number of conjuncts $\geq 16$. The DASL model shows no loss in performance, highlighting better numerical stability. This example highlights the improved numerical stability enabled by DASL which is critical for scaling, especially when learning NN with deeper logical structures.

## A.8 ADDITIONAL EXPERIMENTAL RESULTS FOR VISUAL RELATIONSHIP DETECTION

| | 1% Data | | 5 % Data | |
|---|---|---|---|---|
| Predicate | Baseline | + Knowledge | Baseline | + Knowledge |
| Wear (R) | 0.90 | 1.00 | 0.86 | 1.00 |
| Under (R) | 0.02 | 0.38 | 0.25 | 0.49 |
| Above (R) | 0.30 | 0.37 | 0.44 | 0.58 |
| On | 0.63 | 0.80 | 0.71 | 0.77 |
| Has | 0.10 | 0.07 | 0.18 | 0.14 |
| Next To | 0.15 | 0.06 | 0.16 | 0.15 |

Table 2: Shows class-wise results (Top-1 accuracy) for top performing predicates from the predicate detection task. "Baseline" refers to VGG (baseline) and "+ Knowledge" refers to the knowledge augmented model (see Table 1). (R) means that the predicate was included in one of the rules.

**Class-wise performance:** To provide further insights into VRD, we also report class-wise comparison (Top-1 accuracy) with and without knowledge in Table 2. We see consistent improvements for predicates which were included in one of the rules, *e.g.*, "Wear" and "Under". Interestingly, we also see improvements for the "On" predicate since it has the most training samples and thus the baseline tends to confuse other predicates with "On" (common overfitting issue in DNNs). In such low-data regimes DASL improves performance for "On" by being able to correctly predict related predicates such as "Under" and "Above". We observe some loss in performance for predicates such as "Has" since these are hard to detect and we did not use any rules for them. We also conducted a simple ablation study by removing all the spatial rules (with VGG-SS + Knowledge) to see their impact on performance. We observe a drop in performance for $1\%$ training data (R@100=$81.8$, R@100-ZS=$66.8$) with all rules versus (R@100=$75.3$, R@100-ZS=$58.78$ w/o spatial rules).

| % Data | DASL | [Xie et al. (2019)] | |
| --- | --- | --- | --- |
| | DASL | Only Positive Relations | Additional Negative Relations |
| 1% | 71.5 | 55.5 | 71.0 |
| 5% | 77.7 | 72.4 | 77.6 |

Table 3: Comparison on Visual Relationship Detection with a recent neuro-symbolic based method [Xie et al. (2019)]. We report Top-5 accuracy metric for comparison with their work. Here DASL refers to knowledge augmented model "VGG-SS + Knowledge". We show results with [Xie et al. (2019)] as using both positive and negative object pairs for training. In comparison DASL only uses positive object pairs for training.

We would also like to note that the first-order constraints were always satisfied for VRD since we applied rules as vectors of formulas as discussed in subsection 3.3. Here the formulas are applied directly in conjunction with the output of the VRD and thus by definition the rules are always satisfied. We note that this is different from how the rules were satisfied for MNIST and is made possible by the ability of DASL to extend logical language by allowing formulas to work on output tensors.

**Comparison with [Xie et al. (2019)]:** Here we compare DASL with a recent neuro-symbolic method [Xie et al. (2019)] on the VRD task. This work proposed a graph embedding network that is able to learn semantically-constrained embeddings using propositional formulas and reported improvements on the VRD. The numbers reported in this work are not directly comparable to ours since they used $100\%$ training data, while we use only $1\%$ and $5\%$ training data. Moreover, they reported performance on both positive pairs and negative object pairs (negative pairs are labeled with an extra "no-relation" class). This experimental protocol is different from that commonly used the in literature [Lu et al. (2016)] (also used in our work), which only evaluates on positive pairs. The number of object pairs with negative relationships is quite high ($86\%$ of total boxes), which results in a higher chance performance leading to higher gains on their test set. For a fair comparison we re-trained their model with $1\%$ and $5\%$ data and report Top-5 accuracy on positive object pairs, as done in the literature, in Table 3.

We use the variant "VGG-SS + Knowledge" for comparison since [Xie et al. (2019)] also uses embedding vectors and bounding box coordinates in their model. We see consistent improvements with DASL with the improvements being higher for the $1\%$ case as compared to the $5\%$ case. We also note that [Xie et al. (2019)] uses many more rules (*e.g.*, using 10 spatial relations compared to our 4) than DASL, which gives them a certain advantage. That is because as we showed earlier the performance in $1\%$ data case substantially falls when the spatial rules were removed (R@100=81.8 versus R@100=75.3).

## A.9    IMPACT OF FORMULAS ON TRAINING TIME

We haven't done any specific timing tests, though the runtimes were similar with and without logic. This is theoretically expected as the logic is placed in series (after) the neural networks and is faster to compute. With sufficiently complex logic one could imagine logical computations (and backpropagation) to exceed the computation cost of the neural networks, but we are far from this regime. We also heavily utilize the tensor operations in Pytorch, which allows us to leverage the speedups in training general neural networks.

## A.10    FUTURE WORK

First order logic provides a uniform framework in which we plan to support transfer learning and zero-shot learning by training DASL models on theories where data is abundant and then creating new theories on the same vocabulary that address problems where data is sparse. We also plan to demonstrate the converse capability, training distinct models of a single theory, allowing us to sample models as a technique for capturing true probabilities, similar to Markov Logic Networks [Richardson & Domingos (2006)]. Finally, we are exploring ways to allow DASL to learn rules from data while retaining explainability and integrating smoothly with user defined logic.

