# OpenReview forum: "DEEP ADAPTIVE SEMANTIC LOGIC (DASL): COMPILING DECLARATIVE KNOWLEDGE INTO DEEP NEURAL NETWORKS"
_ICLR.cc/2021/Conference — Reject_

### Official Review · AnonReviewer2 · 2020-10-28
**Deep adaptive semantic logic as a framework for knowledge-driven neural network design**

**Rating:** 5
**Confidence:** 4

**Review:**

This paper introduces Deep Adaptive Semantic Logic (DASL) as a novel framework that automates the generation of deep neural networks on the basis of some background domain knowledge. A first-order logic formalism (which exploits Tarski models) is used to represent knowledge, and, subsequently, to model deep networks on the basis of such knowledge. This resembles ideas coming from the neuro-symbolic community, such as the pioneering work by Towel and Shavlik on KBANNs, and more recent approaches such as Neural Tensor Networks, DeepProblog, Lyrics.

The model is evaluated on a toy problem (extracted from MNIST) and the task of visual relationship detection, that aims to predict relations between objects in the scene.

Although the work is interesting, and perfectly in scope with the conference, focusing on a very hot topic in AI, I have to raise some concerns regarding the experimental evaluation, which basically does not compare against similar approaches, nor it considers similar tasks in the literature.

In particular, no comparison is presented with other approaches from the Neuro-Symbolic community: more precisely, maybe some tasks could have been borrowed from previous works. For example, the DeepProbLog paper presents a similar task regarding sums with MNIST digits? Or, in the setting of image classification, the framework of "Learning from constraints" has been evaluated on a task which considers exploiting class hierarchies and commonsense (e.g., see "Image Classification Using Deep Learning and Prior Knowledge", AAAI Workshops, 2018).

Other comments:

* What is the state-of-the-art for the VRD task? Is DASL competitive?

* The way references are used in the text should be made consistent. In Sections 1 and 2, there is no parentheses before references (e.g., "...estimated from data LeCun et al. (2015)") whereas in Section 3 there are parentheses, which is to be preferred (e.g., "...by boolean Tarski models [Chang & Keisler (1973), Weiss & D’Mello (1997)]").

- "techniques such conditional random fields" -> "techniques such as conditional random fields"
- "data scarce conditions" -> "conditions of data scarcity"

---

> ### Author Response · Authors · 2020-11-13
> **Explain comparison with prior neuro-symbolic methods on MNIST and provide additional comparison on VRD**
>
> We thank the reviewer for finding our work interesting and its focus on a very relevant topic in AI.
>
> **Missing comparison against similar approaches**.
> We did compare our approach with two representative neuro-symbolic methods in *Section A.7-Appendix* (referenced on page 2)-- *Serafini & Garcez (2016)* and *Rocktäschel (2015)* —both of which are based on using product-t norms. Compared to DASL’s logit representation, they use either probabilities or log-probabilities to represent the truth values. We demonstrated that both of these techniques fail catastrophically with increasing the number of conjunctions in logic. This highlights one of the core strengths of DASL which is the ability to operate with deeper logical structures with improved numerical stability. We agree with reviewer’s concern that this comparison might have been missed and will add discussion around this in the paper.
>
> As done for *AnonReview3*, we compare our approach with a recent neuro-symbolic method *Xie et al. (2019)* that uses first-order-logic for VRD problem. We observe consistent improvements compared to their method especially for 1% training data. We provide the result table here and refer to the review for *AnonReview3* for more details.
>
> |---%---|DASL--|Xie et al.|Xie et al. with additional|.
> |-Data-|---------|------------|---negative relations-----|.
> |--------|---------|------------|----------------------------------|.
> |--1%--|--71.5--|---55.5---|------------71.0----------------|.
> |--5%--|--77.7--|---72.4---|------------77.6----------------|.
> Metric is top-5 accuracy as used in *Xie et al. (2019)*
>
> Our MNIST example is presented as a toy problem intended solely for the pedagogical purpose of illustrating how the use of knowledge reduces the need for data in our framework. The *DeepProbLog* paper is indeed similar in that it demonstrates arithmetic operations on images of digits, but the method is so different from ours that a comparison would do little to improve the illustration.  Whereas DASL creates a semantic interpretation of the arithmetic rules within which they inherently hold true, *DeepProbLog* takes a syntactic approach, compiling the rules into a Bayesian model that effectively executes them.
>
> In essence DASL builds NN layers that leverage standard backpropagation for reasoning, fully integrating logic into neural networks using back propagation as the single solution technique, while DeepProbLog integrates two distinct solvers (backprop and ProbLog). These different techniques do not, necessarily, result in the same outputs, which the MNIST example addresses by showing the ability of DASL to solve modulo addition problems, in a way that is distinct from how DeepProbLog works.
>
> Other comments:
> - What is the SOTA for VRD task: We do compare our model with a SOTA method *Liang et al. (2018)* in our paper. However, we would like to re-iterate that our intention was to show that knowledge can improve learning significantly when only small amounts of data (1% and 5%) are available. Therefore, a direct comparison with recent methods, that rely on using 100% data, is not helpful.
> - As recommended, we have corrected all the references with parentheses.
> - Thanks for catching the typos. We have corrected those.
>
> We thank them for their valuable comments and will update the main text and Appendix with these discussions.

---

> > ### Comment · AnonReviewer2 · 2020-11-23
> > **Thanks**
> >
> > I thank the authors for their clarifications.

---

### Official Review · AnonReviewer1 · 2020-10-28
**Unifies logical and neural network models. Seems like a good paper that can be stronger by clarifying some details.**

**Rating:** 6
**Confidence:** 3

**Review:**

The paper describes an approach to combine first-order knowledge bases with deep networks. The idea is to represent logical operators as neural components that produce continuous values (using a logistic output) instead of boolean values. The proposed approach is formalized using Tarski models. Here, it is shown that if we consider each parameterization as a model, the learning samples models from the logical theory. Results are proven that such an approach is both sound and complete for any first-order logic theory.

Neuro-symbolic models seem very promising and therefore the paper addresses a very significant problem. The approach seems similar to Rockstachel et al. 2017’s approach for theorem proving in first-order logic. It seems like the main contribution though is that their model is more general and they prove soundness and completeness of their approach. Maybe a little bit more on why the approach is more general than existing approaches can help motivate the novelty aspect more clearly.


The experiments seem to be well thought-out though only 1 real domain and 1 toy domain are explored. However, I was not sure how the knowledge bases are constructed for the VRD problems. The single example seems clear. But how is it done for any problem in general? What types of logical formulas were pre-specified corresponding to visual relationships since I imagine this can be quite extensive. Is there any impact on training time/effort as the formulas become more complex since the architecture of the network becomes more complex. Also, does the expert knowledge change for each image? I think some more details on this task will help clarify the experiments and make the results more easily reproducible.

In summary, this seems like a nice paper but a little fuzzy on the details of the experiments which can be clarified better.

---

> ### Author Response · Authors · 2020-11-13
> **Explain how we generalize prior work and create knowledge bases; impact of formulas on training time**
>
> We thank for reviewer for noting that our approach is sound and complete for any first-order logic, appreciating the significance of the problem and promising aspect of this work.
>
> **Why our approach is more general**.
> We especially thank this reviewer for pointing out that we have not made clear the sense in which our approach is "more general” than previous approaches.  We will correct this in the paper.  The phrase "first order logic” is often applied to any fragment of first order logic that goes beyond propositional logic by including predicates with arguments.  However, there are many such formalisms with varying degrees of expressiveness and varying difficulty for reasoning.  The point of the soundness and completeness proofs is to demonstrate that DASL captures reasoning in full first-order logic with equality, as defined, for example, by Tarski-style semantics; this is the most expressive (hence, “most general”) first-order language, subsuming sql, sparql, prolog, datalog, and OWL, to name a few common examples.  Many neuro-symbolic approaches provide examples which are quantifier-free, function-free, and equality-free, without indicating the exact scope of the language addressed.  *Rocktäschel et al. (2017)*, as pointed out by the reviewer, demonstrate theorem-proving for knowledge base queries in prolog.  In prolog, one can write “the father of the father is the grandfather”:
> grandfather(x, z) $\gets$ (father(x, y) $\wedge$ father(y, z)).
> but one cannot write “everyone has a father”, which is written in full first-order logic as:
> ($\forall$x)($\exists$y) father(y, x).
> Prolog does not allow for nested quantifiers at all, whereas full first order logic allows arbitrary quantifier nesting.  Also, prolog uses “negation as failure”, also known as the “closed world assumption”.  This means that prolog answers every yes/no question with yes or no, whereas full first order logic can answer yes, no, or that we can’t tell from the given information.  In a knowledge base consisting only of the facts “Andy is the father of Bill” and “Bill is the father of Chris”, the prolog answer to the query “who is the father of Andy” is “no one”, because Andy’s father is not made explicit in the knowledge base, whereas in full first order logic the answer is “can’t tell”.  DeepProbLog also restricts itself to the prolog language, so in that sense it is narrower than DASL.
>
> We would also like to point out the DASL is also better at scaling with depth of the logic compared to *Rocktäschel et al. (2015)*. In *Section A.7-Appendix*) we show that for training with negated conjunctions, *Rocktäschel et al. (2015)* fails catastrophically for assertions with 16 or more conjuncts, while DASL has no such issues due to higher numerical stability.
>
> **How the knowledge base was constructed for VRD**.
> We would like to note two approaches for building such knowledge bases. The first method is a manual method where a user can supply logical rules that are focused on classes where the model does not work well. The second method is to exploit knowledge bases such as Concept-Net, WordNet or meta-data of prior datasets such as Visual Genome to build such knowledge bases. For VRD, we started with Visual Genome to build our knowledge base and then kept a subset of rules which we found to be most helpful (see *Section A.6-Appendix*). While building these rules we found it helpful to group the predicates in a rule set. For example, predicates such as "Above”, "Over” etc. were part of "above” rule set which checked whether the subject is above the object. We understand the concern regarding the feasibility of constructing a knowledge base but in our view there is sufficient work in the literature showing how machine learning algorithms have exploited knowledge bases from pre-existing resources [1][2].
>
>
>
> **Impact of formulas on training time**.
> We haven’t done any specific timing tests, though the runtimes were similar with and without logic. This is theoretically expected as the logic is placed in series (after) the neural networks and is faster to compute. With sufficiently complex logic one could imagine logical computations (and backprop) to exceed the computation cost of the neural networks, but we are far from this regime. We also heavily utilize the tensor operations in Pytorch which allows us to maintain the speedup in training general neural networks.
>
> **Does expert knowledge change for each image?**.
> No, the logic is fixed for all images, though the data (including object types) change.
>
> We again thank the reviewer and will add these points to the Appendix.
>
> 1. Aditya, Somak, et al. "Combining Knowledge and Reasoning through Probabilistic Soft Logic for Image Puzzle Solving." UAI. 2018.
> 2. Moldovan, Dan I., and Vasile Rus. "Logic form transformation of wordnet and its applicability to question answering."ACL.2001.

---

### Official Review · AnonReviewer3 · 2020-10-29
**Interesting work but lack of empirical comparison and contrast with prior work**

**Rating:** 3
**Confidence:** 5

**Review:**

This paper focuses on the problem of augmenting the user-provided knowledge into Neural networks. The core premise of the work is that the existing approaches tend to replace the True and False with values in [0,1] and then use maximum/minimum operators which have vanishing gradients. The key idea is to use logit representation and therefore, the authors hope to avoid the vanishing gradients problem. The paper presents empirical results to argue that embedding knowledge helps.

I think its an interesting paper but somehow the comparison with prior work is not properly contrasted, which makes me believe that the paper has probably reinvented quite a bit of aspects. An important missing work is [1], which in fact uses product and addition instead of minimum/maximum.  (It is well known that exists and forall operators can be expressed using disjunction and conjunction)

Also, Xie et al (NeurIPS 2020) build on [1], and  do perform empirical on the same VRD dataset and achieve significant improvements; their numbers seem better than what is reported in Table 1.

In light of the above remarks, it is hard to understand the new contribution of the paper.


[1] https://arxiv.org/abs/1711.11157 (Published at ICML 2018)

---

> ### Author Response · Authors · 2020-11-13
> **Logit contribution clarified, missed contributions highlighted, and improvements over Xie et al. made explicit**
>
> We thank the reviewer for their comments.
>
> **Comparison with Prior Work [1]**.
> The logit representation is the only contribution discussed by the reviewer, but we also believe our work substantially advances over the state-of-the-art by introducing model-theoretic completeness proofs that provide a framework for exploring properties of neuro-symbolic systems in general (also noted by *AnonReviewer1*), and we believe that the extensions of the language will be useful in generalizing the kinds of knowledge that get developed for neuro-symbolic systems.
>
> Additionally, it seems that we did not communicate the logit contribution to this reviewer. We certainly do not claim to be the first to use the product of truth values for conjunction; in fact we mention product in the same sentence in which we first mention maximum/minimum, and it is only product that we call out for vanishing gradients. We cite *Rocktäschel (2015)*, *Serafini & Garcez (2016)*, both of whom use product.  We compare to these methods because they are most similar to our work, and the logit representation allows us to learn with larger conjunctions (more complex logic and more data points) than previous representations. In *Section A.7-Appendix* (referenced on page-2) we show both *Rocktäschel (2015)* and *Serafini & Garcez (2016)* fail catastrophically for learning with higher numbers of conjuncts. We will include a brief description in the paper.
>
> The reviewer’s reference [1] introduces a novel "semantic loss” function for training propositional valuation functions.  This loss function is interesting and unique, but very different from our loss function, which is much more similar to *Rocktäschel (2015)* and *Serafini & Garcez (2016)*.  Also similar to these earlier neuro-symbolic works is our direct representation of the predicates of the language.  While [1] does follow these earlier works in implementation of the Boolean connectives, it is restricted to propositional logic, which handles very different types of problems than those the neuro-symbolic community addresses with predicate logic.  Propositional logic, for example, cannot express "For any subject s and object o, if s is riding o, then o is ridable and s is capable of riding” (as we use in *Figure 1*).  We will clarify this in the related work section.
>
> **Comparison with Xie et al.**.
> The reviewer correctly points out that *Xie et al* produce higher accuracy on VRD. However, the results are not comparable to ours since *Xie et al* reported results with 100% training data, while we use only 1% and 5% training data. *Lu (2016)* and *Yu (2017)* also train on all training data and achieve a higher score than do we.  Our intention was to show that knowledge can improve learning significantly when only small amounts of data are available. We found from the code released by * Xie et al.* that they report top-5 accuracy on both positive pairs and negative object pairs (negative pairs are labeled with an extra “no-relation” class). The metric used by *Xie et al.* is not comparable to that commonly used in the literature *Lu (2016)*, which we used.  The number of object pairs with negative relationships is quite high (86% of total boxes), which results in a higher chance performance leading to higher gains on their test set.
>
> Based on the reviewer’s suggestion, we did experiment with *Xie et al.* by training their approach with 1% and 5% data and reporting top-5% accuracy on positive object pairs as done in the literature. We compare with the variant “VGG-SS + Knowledge” since *Xie et al.* also uses embedding vectors and bounding box coordinates in their model. We see consistent improvements with DASL with the improvements being higher for 1% case as compared to 5% case. We would also like to note here that *Xie et al.* uses more rules (e.g. spatial constraints over 10 spatial relations compared to 4 of ours) as compared to DASL, which gives them a certain advantage. That is because as we showed earlier the performance in 1% data case substantially falls when the spatial rules are removed (R@100=81.8 vs. R@100=75.3).  We thank the reviewer for this suggestion and will add this discussion in the paper.
>
> |---%---|DASL--|Xie et al.|Xie et al. with additional|.
> |-Data-|---------|------------|---negative relations-----|.
> |--------|---------|------------|----------------------------------|.
> |--1%--|--71.5--|---55.5---|------------71.0----------------|.
> |--5%--|--77.7--|---72.4---|------------77.6----------------|.
> Metric is top-5 accuracy.
>
>
> 1. Xu, Jingyi, et al. "A semantic loss function for deep learning with symbolic knowledge." ICML. 2018.
> 2. Xie, Yaqi, et al. "Embedding Symbolic Knowledge into Deep Networks." NeurIPS. 2019.

---

> > ### Comment · AnonReviewer3 · 2020-11-23
> > **Need clarifications on results**
> >
> > Apologies for late response.
> > One thing that is clear to me if there is a reason to believe that training data with negative relations is expensive to compute. Essentially, the results seem on line with Xie et al, so do we have a  strong argument for Xie et al without negative negative relations.

---

> > > ### Author Response · Authors · 2020-11-23
> > > **Clarification regarding comparison with Xie et al.**
> > >
> > > We thank the reviewer for their comment and suggestions. We do not believe that negative relations are expensive to compute and did not mean to imply that in our response. Furthermore, we would like to first mention that *Xie et al.* used a **non-standard evaluation** protocol for evaluating the predicate detection task. As we mentioned earlier the use of negative relation tends to falsely inflate the performance in general (86% relations are negative). Moreover, the vrd network can be trained with only positive relations with a softmax classifier.  We agree that negative relations can be used for training. However, that is very different from our approach. We use the same test set (without negative relations) that is **standard in the literature** (*Lu et al. (2016)*) to compare our work to that of Xie et al.  Comparing different approaches on different test sets that have different statistics does not result in a meaningful comparison.
> > >
> > > We do in fact have a strong argument for improvement over *Xie et al.* w/o negative relations where we show that our model outperforms *Xie et al*. by a margin of 16% (absolute) with 1% training data. Additionally, we also outperform *Xie et al.* even when they use additional negative relations for training.
> > >
> > > 1. Lu, Cewu, et al. "Visual relationship detection with language priors." European conference on computer vision. Springer, Cham, 2016.

---

### Official Review · AnonReviewer4 · 2020-11-01
**An interesting paper**

**Rating:** 5
**Confidence:** 5

**Review:**

The paper integrates a neural encoding of first-order logic with deep learning architectures, supplementing training data with declarative knowledge. The approach is experimentally evaluated on MNIST and visual predicate detection, demonstrating  a reduction in data requirements.

The paper deals with an important topic and provides very promising results. However, I would've liked to see a more extensive experimental evaluation. Also, which performance gain does additional knowledge produce for visual predicate detection, compared to the current SOTA for that task. So, are the first-order constraints always satisfied by the produced results?

After rebuttal: The long discussion with the authors to clarify the questions in my review and further related questions actually shows that the paper is not really clear and would still benefit from a substantially improved presentation. So, I slightly downgraded my overall rating of the paper.

---

> ### Author Response · Authors · 2020-11-13
> **Provide additional experimental evaluation on VRD and explain constraint satisfaction**
>
> We thank the reviewer for their helpful comments and appreciation of the problem and our results.
>
> **Additional Experimental Details**:
> We understand your request for additional details about the experimental evaluation. First, as shown in *Table 1*, the improvements with DASL were higher for the zero-shot setting (R@100 in Zero-shot was 59.4 with VGG and 70.1 with DASL). This is expected because knowledge helps the most when the training data is minimal since the deep network will not generalize well in such cases. To provide further insights into VRD, we also report class-wise comparison (top-1 accuracy) with and without knowledge (using “VGG”) below.
>
>
> | Predicate|----------1% Data-------|----------5% Data-----|.
> |--------------|-----------|---------------|-----------|--------------|.
> |--------------|--DASL--|-Baseline--|-DASL---|-Baseline--|.
> |--------------|-----------|---------------|---------   |--------------|.
> |--Wear(R)-|--1.00---|-----0.90-----|--1.00---|--0.86------|.
> |-Under(R)-|--0.38---|-----0.02-----|--0.49---|--0.25------|.
> |-Above(R)-|--0.37---|-----0.30-----|--0.58---|--0.44------|.
> |--On---------|--0.80---|-----0.63-----|--0.77---|--0.71------|.
> |--Has--------|--0.07---|-----0.10-----|--0.14---|--0.18------|.
> |--Next To---|--0.06---|-----0.15-----|--0.15---|--0.16------|.
>
>
> (R) means that the predicate was included in one of the rules. We see consistent improvements for predicates which were supplied with a rule e.g. "wear", "under". Interestingly, we also see improvements for "On" predicate since it has the most training samples and thus the baseline tends to confuse other predicates with "On" (common overfitting issue in DNNs). In such low-data regimes DASL improves performance for "On” by being able to correctly predict related predicates such as "Under” and "Above”. We observe some loss in performance for predicates such as "Has" since these are hard to detect and we did not use any rules for them.
>
> We also conducted a simple ablation study by removing all the spatial rules (with "VGG-SS + Knowledge") to see their impact on performance. We observe a drop in performance for 1% training data (R@100=81.8, R@100@ZS=66.8 with all rules vs. R@100=75.3, R@100@ZS=58.78 w/o spatial rules).
>
> We shall add this discussion in the paper and the table to the Appendix. We appreciate these comments as it improves our empirical analysis.
>
> **About First-Order Constraints**:
> Yes, the first-order constraints were always satisfied for VRD since we applied rules as vectors of formulas as discussed in *Subsection 3.3*. Here the formulas are applied directly in conjunction with the output of the VRD (shown below) and thus by definition the rules are always satisfied. We note that this is different from how the rules were satisfied for MNIST and is made possible by the ability of DASL to extend logical language by allowing formulas to work on output tensors.
>
> \begin{align}
> 	&(\forall (\mathbf{I}, s, o, y):\mathbf{D_\mathrm{vrd}}) [\pi_y(\mathrm{vrd}(\mathbf{I}, s, o) \newline
> 	& \quad \wedge (\mathbf{h}_{\mathrm{Riding}} \rightarrow \mathrm{CanRide}(s) \newline
> 	& \qquad \qquad \quad \quad \wedge \mathrm{Ridable}(o)))]
> \end{align}

---

> > ### Comment · AnonReviewer4 · 2020-11-20
> > **Please clarify:**
> >
> > Many thanks for your answer; however, it is still unclear to me how you can give a guarantee that the constraints will be always satisfied (as the background knowledge is only encoded in the loss).
> >
> > For example, in the MNIST example, how can the authors guarantee that the neural network will never output a triplet like (3, 5, 9)? (When we have the constraint stating that “if X_1 is 3 and X_2 is 5, then X_3 must be 8”).
> >
> > Similarly, for the VRD example, how can the authors guarantee that if the truth value states that the predicate “Riding” is true, then the neural network will also predict as true the predicates “Ridable” and “CanRide”?
> >
> > Could you please clarify?

---

> > > ### Author Response · Authors · 2020-11-20
> > > **Clarification regarding satisfaction of constraints**
> > >
> > > We again thank the reviewer for their comment. We misunderstood your question earlier and only answered it in our specific approach for VRD. For a general problem where the knowledge is encoded using our loss function (as is done for MNIST), there is no guarantee that the rules will be satisfied. When the training loss is small, the network approximately satisfies the rules for the training data.  When the network is applied to data outside the training set, generalization is not guaranteed, just like for any learning problem.
> > >
> > > For our VRD approach specifically, the subject and object classes are explicitly provided as inputs (they are not detected by a learned network), so the consequent of each implication is computed explicitly as exactly 0 or 1, and the antecedent is computed accordingly.  These rules will therefore always be satisfied because they are deterministic.  In this architecture, the outputs of the classification network are not fed into the rules; rather, some of the outputs are masked by the rules when appropriate (*Section 4.2*).
> > >
> > > We agree with the reviewer that in the general VRD case the satisfaction of our rules cannot be guaranteed especially when the truth about the objects being “Ridable” and “CanRide” is being predicted through separate (learnable) neural network. In our specific case however, the truth values for these were already available as the subject and the object are known for predicate detection.

---

> > > > ### Comment · AnonReviewer4 · 2020-11-22
> > > > **A related question:**
> > > >
> > > > Thanks you for your answer. I have another question: Since the introduced rule “reduces the learning burden of the classifier for “Riding” class by allowing feedback only when CanRide(s) is true”, it is possible that it causes the generation of many False Positives. Thus, could you please also report the Precision@50 and Precision@100?

---

> > > > > ### Author Response · Authors · 2020-11-22
> > > > > **Reporting Precision@50 and Precision@100**
> > > > >
> > > > > We thank the reviewer for their comment. We should clarify that the rules remain in place during testing, so false positives will not be generated for cases where the rules apply. We see below that DASL does not introduce more false positives than the baseline, but we should also point out that the original authors (*Lu et al.(2016)*) selected Recall compared to Precision because, for large-scale crowd-sourced datasets such as Visual Genome, it is often difficult to exhaustively label all possible relationships for an object pair. For example, both "On” and "Above” would be reasonable to describe object pair "Person”-"Road”, but only one of those labels will be provided as the correct annotation, so precision scores (for any approach) will be lower than they would if all applicable labels were provided in the ground truth. We have provided the Precision@50 and Precision@100 for 1% and 5% data below (averaged over 20 runs) as requested. The Baseline is "VGG (baseline)” and DASL is "VGG (baseline) + Knowledge”.
> > > > >
> > > > >
> > > > > | -------------|----------------1% Data---------------------|-----------------5% Data---------------------|.
> > > > > |--------------|--Precision@50--|--Precision@100--|--Precision@50--|---Precision@100--|.
> > > > > |-Baseline-|-------9.6 +- 1.3----|------6.1 +- 0.7------|------11.3 +- 0.1----|------6.8 +- 0.0------|.
> > > > > |----DASL---|-----10.8 +- 0.1----|------6.6 +- 0.4------|------11.8 +- 0.1----|------7.0 +- 0.0------|.
> > > > > (p-value < 0.005 in all cases; Reporting mean with standard deviation).
> > > > >
> > > > >
> > > > > We see that DASL still shows improvement over the baseline based on the Precision@K metric. The trends are similar to recall where the improvements are higher for 1% data (+1.2 with Precision@50) as compared to 5% data (+0.5 with Precision@50). We note that the numbers are lower for Precision@100 compared to Precision@50 since the former considers more boxes for computing the metrics.
> > > > >
> > > > > 1. Lu, Cewu, et al. "Visual relationship detection with language priors." European conference on computer vision. Springer, Cham, 2016.

---

> > ### Comment · AnonReviewer4 · 2020-11-23
> > **Another question**
> >
> > Thanks for your answer.
> >
> > One more question regarding the VRD experiment, where the neural network vrd(I, s, o) is used. Such network outputs raw scores for predicate classes, where I is the input RGB image and s and o are the indices of the subject and object classes respectively. In the rule though, h_Cls is used. h_Cls is a one-hot vector of truth values, which is True at the index of the predicate class “Cls” and False elsewhere. Since these rules are deployed also at test time, does it mean that part of the ground truth annotation is used at test time? (i.e., In order to use this framework, for every image in the test set, will the authors need to know whether s is "riding" o or not? --> if the predicate "riding" is true or not? ).

---

> > > ### Author Response · Authors · 2020-11-23
> > > **Clarification regarding h_CLs**
> > >
> > > We thank the reviewer for their comments. We would like to clarify that these rules **do not use any information about ground-truth** either during training or during testing. The reviewer correctly describes h_Cls, but this is not a function of the output class; we are defining the vector h_Cls for each class for which we write rules. But every rule applies to every image.  Thus, there are rules for h_Riding which are applied to every image, and there is a rule for h_Above which is applied to every image, and there is a rule for h_Wear which is applied to every image, etc.  (Some predictaes have multiple rules, some have one rule, some have no rules at all.).
> > >
> > > As mentioned in our previous comment, the purpose of these rules is to prevent the model from predicting certain predicates based on whether they satisfy some high-level conditions. For example, in the case of “Riding” predicate, the rules will only let the model predict "Riding” when the subject "CanRide” and the object is "Ridable”. Another example is "Above” predicate, where the condition is that the subject should be above the object. These rules are universal (except in rare cases) and do not depend on the ground-truth in any way.
> > >
> > > Let’s work through a concrete example.  Suppose that there are only 3 possible outputs (just so we don’t need an example with 70 values): "Riding”, "On”, and "Wearing” (in that order). Then h_riding is the vector (1, 0, 0), h_on is the vector (0, 1, 0), and h_wearing is the vector (0, 0, 1). Consider two images Image1 and Image2.  Suppose that for both images, vrd(I, s, o) outputs the 3 scores (23, 17, 9) for these classes, prior to the softmax layer.  For Image1, suppose that CanRide(s) and Ridable(o) are both true (recall that we are given the classes for s and o as inputs, so these determinations do not require learned functions).  Then the rule h_riding $\rightarrow$ CanRide(s) $\wedge$ Ridable(o) applies the $\wedge$ operator to (1, 0, 0) on the left and (1) on the right.  DASL automatically broadcasts, so the output of this operator is (1, 1, 1) based on the truth table for $\rightarrow$. Let’s suppose that the "On" and "Wearing" rules also output (1, 1, 1) (suppose that there was one rule for each predicate).  In the next step, (following the rule on page 8), the $\wedge$ operator is applied to vrd(I, s, o) and all three rule outputs.   $\wedge$ ( (23, 17, 9), (1,1,1), (1,1,1), (1,1,1) ) = (23, 17, 9).  This is now passed to the softmax (in the $\pi$ operator), and the prediction will be for Riding.  We don’t yet know whether that was the correct answer.
> > >
> > > For Image2, suppose that CanRide(s) again is true but that this time Ridable(o) is false.  Then CanRide(s) $\wedge$ Ridable(o) evaluates to 0, h_ridable $\rightarrow$ 0 evaluates to (0, 1, 1).  Suppose that the "On" and "Wearing" rules again evaluate to (1, 1, 1).  Then $\wedge$( (23, 17, 9), (0, 1, 1), (1, 1, 1), (1, 1, 1) ) = (0, 17, 9).  This triple now goes to softmax and this time prediction will be for "On".  Note that in both cases, all rules were applied regardless of the correct output class and note that even though the vrd network prefers Riding in Image2, that is only a component of the full prediction system, and that system has suppressed the Riding prediction. We are able to apply such rules due to DASL’s ability to work with tensors of Truth values (*Section 3.3*). Moreover, as previously shown, learning vrd network with these rules is able to improve its performance for data scarce settings (1% and 5% training data).
> > >
> > >
> > > Truth-Table for Implication (A is antecedent and B is consequent).
> > >
> > > A$\rightarrow$B.
> > > A|B|Output.
> > > T|T|T.
> > > T|F|F.
> > > F|T|T.
> > > F|F|T.

---

### Author Response · Authors · 2020-11-21
**Updated the submission based on reviewer's comments**

We thank the reviewers for their comments. We have updated the manuscript with the following changes based on the comments:


1. [AnonReviewer3, AnonReviewer2] Added a section on "Comparison with related neuro-symbolic" methods in the *main text*. It provides comparison with Serafini & Garcez (2016), Rocktäschel (2015), and Xie et al.(2019). Details are provided in the *Appendix*.

2. [AnonReviewer3] Provided additional details about contributions over prior works and differences wrt. [1] ("Xu et al.(2018") in the *Related Work* section.

3. [AnonReviewer4] Provided additional experimental details for VRD in the *Appendix* as well as the *Experiments* section.

3. [AnonReviewer1] Added discussion on why our approach is more general in the *Approach* section.

4. [AnonReviewer1] Added discussion on knowledge bases and impact of formula on training time in the *Appendix*.

5. [AnonReviewer2] Corrected typos and citations.

6. During the rebuttal, we found that we posted a wrong version of the graph for *Figure 3* (in Appendix). We now updated the graph based on the corrected version of the code.

7. We also corrected some additional typos in the *main-text* and the *Appendix*.

Please **note** that we have added the *Appendix* at the end of *main text* now.

---

### Decision · Program_Chairs · 2021-01-07
**Final Decision**

**Decision:**

Reject

**Comment:**

The reviewers and AC appreciate the improvements made to the paper and thank the authors for engaging with the reviewer questions.
There are now quite a few neuro-symbolic approaches, and they are all rather similar. This places a larger burden on the authors to have a thorough and systematic experimental comparison and related work discussion. Reviewers also believe the clarity of the paper should still be improved. The revised paper already made good progress in addressing these concerns, yet the reviewers still believe the paper would strongly benefit from another round of revisions.